# Prothrombinase processivity is conferred by substrate allostery

Fatma Işık Üstok [1,3], Alexandre Faille[1,2,3], Alan J Warren [1,2] & James A Huntington [1✉]

## Abstract

The prothrombinase complex, comprised of factor (f) Xa and fVa, converts prothrombin to thrombin through sequential cleavage at two sites in a rapid and processive manner. The molecular basis of prothrombin processing is an enzymatical mystery that to solve requires structural insight into how the substrate and intermediate bind to prothrombinase. Here we present two 3.1 Å cryo-EM structures of prothrombinase bound to prothrombin and to meizothrombin. The prothrombin complex revealed a surprising interaction between the end of the heavy chain of fVa with exosite I of prothrombin, accounting for 70% of the contact interface. Triggering of the zymogen-to-protease conformational change following cleavage at Arg320 alters all domain-domain and fVa interactions observed for prothrombin, and results in a large-scale rearrangement of meizothrombin that presents the second cleavage site (Arg271) for processing. Together, these structures reveal a remarkable enzymatic mechanism that requires the active participation of the substrate itself, and introduces a new paradigm of 'substrate allostery'.

Subject Categories Post-translational Modifications & Proteolysis; Structural Biology

## Introduction

The serine protease thrombin is the enzyme that clots blood; it is the only endogenous factor capable of cleaving fibrinogen into fibrin and is the most potent activator of platelets (Lane et al, 2005). Inappropriate or excessive thrombin generation is the cause of all forms of thrombosis, and insufficient thrombin generation results in bleeding (Mann et al, 2003). Its zymogen precursor prothrombin is comprised of a gamma carboxyglutamic acid (Gla) domain, two kringle (K) domains and a serine protease (SP) domain, with the Gla-K1 domains and the K2-SP domains each forming rigid units separated by a flexible 26-residue linker (Davie and Kulman, 2006)

(Fig. EV1A). Prothrombin circulates predominantly in a closed configuration with the Gla-K1 unit (also known as fragment 1 or F1) interacting with the SP domain (Pozzi et al, 2016; Pozzi et al, 2013). Conversion of prothrombin to thrombin requires proteolytic cleavage at two sites, after Arg271 and Arg320 (Fig. EV1B), by the enzyme complex prothrombinase, comprised of a serine protease, factor (f) Xa, and a large cofactor, fVa (Krishnaswamy, 2013; Mann, 2021). To limit thrombin production to sites of vascular damage, prothrombinase can only assemble on activated phospholipid (PL) surfaces that are rich in phosphatidylserine (Krishnaswamy, 1990; Skogen et al, 1984). Factor Xa on its own can cleave prothrombin at both sites, but at rates too slow to be of physiological relevance and in an order (Arg271 first) that separates the membrane-binding Gla domain and two K domains (F1.2) from the zymogen form of the SP domain (Prethrombin-2 or Pre-2). Assembly of prothrombinase accelerates cleavage of prothrombin by ~500,000-fold and enforces initial cleavage at Arg320, producing the active, membrane-anchored form meizothrombin as the intermediate. Each prothrombinase complex can convert over 100 molecules of prothrombin to thrombin per second in a processive manner, without apparent dissociation and reassociation of the intermediate (Krishnaswamy, 2013). How efficiency and processivity are conferred by the cofactor fVa lies at the heart of the process of blood coagulation and remains largely unresolved.

Factor Xa is comprised of an N-terminal Gla domain, two epidermal growth factor-like (EGF) domains and an SP domain (Jackson, 2021). The Gla and EGF1 domains associate into an intimate unit connected to the EGF2 domain by a 7-residue linker. The EGF2 domain forms extensive non-covalent contacts with the SP domain, which are also connected by a disulfide bond. Factor Xa is fully active against peptidyl substrates and is not allosterically activated by fVa binding. Factor V circulates as a procofactor comprised of three A domains (~320 residues each), an unstructured B-domain of 889 residues and two small membrane-binding C domains (155 residues each) (Schreuder et al, 2019). The presence of the B-domain renders the procofactor unable to bind to fXa or to prothrombin. Activation to the two-chain fVa is achieved through excision of the B-domain by thrombin or fXa cleavage after Arg709 and Arg1545 (A1-A2 is the heavy chain and A3-C1-C2 is the light chain).

We previously created a variant of human fXa with 17 mutations (M17) to the EGF2 and SP domains that conferred high affinity for human fVa in the absence of PL, and which together recapitulated

[1]Cambridge Institute for Medical Research, Department of Haematology, University of Cambridge, The Keith Peters Building, Hills Road, Cambridge CB2 0XY, United Kingdom. [2]Stem Cell Institute, University of Cambridge, Cambridge CB2 0AW, United Kingdom. [3]These authors contributed equally: Fatma Işık Üstok, Alexandre Faille. ✉E-mail: jah52@cam.ac.uk

# Side view

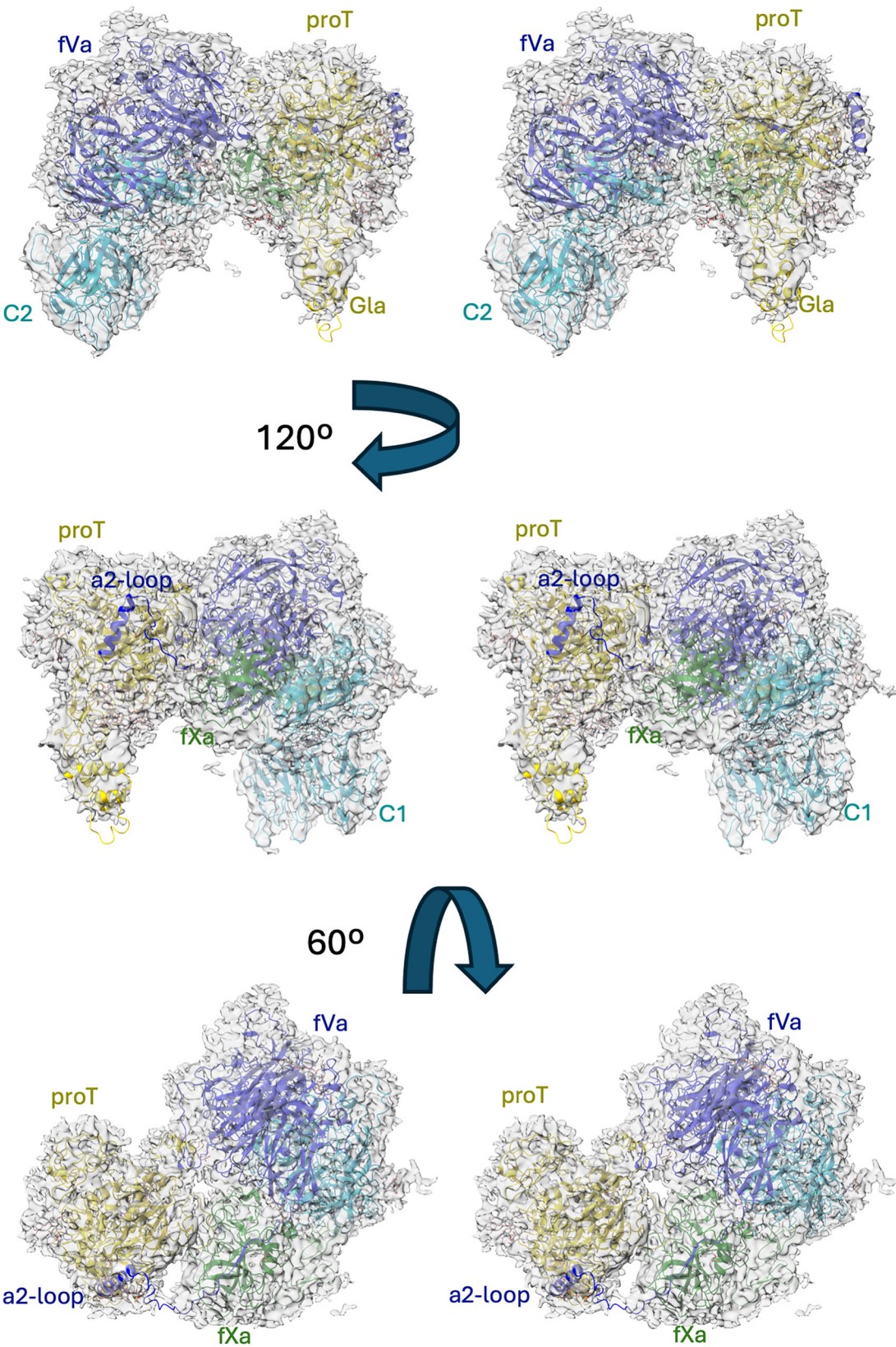

**Figure 1.   Stereo views of the structure of the prothrombinase-prothrombin complex with cryo-EM map.**

Three views of the final coordinates of fVa, fXa and prothrombin with surounding map are shown. The heavy chain of fVa (A1-A2 domains, including the a2-loop) is colored blue; the light chain of fVa (A3-C1-C2 domains) is colored light blue; the EGF2 and SP domains of the fXa are colored light green and green, respectively; prothrombin (proT) is colored yellow; glycosylation is indicated by gray sticks. Molecules and certain domains are labeled for clarity, and the figure was made using ChimeraX.

the prothrombin processing activity of fully-assembled wild-type prothrombinase (Ustok and Huntington, 2026). We then used the M17-fVa complex to determine the structure of prothrombinase by cryogenic electron microscopy (cryo-EM) (Ustok et al, 2026). It revealed an extensive interface involving the A2 and A3 domains of fVa and the SP and EGF2 domains of fXa, driven in large part through electrostatic complementarity. One-third of the interface was contributed by the unstructured region that protrudes from the C-terminus of the A2 domain to the cleavage-activation site at Arg709 (657–709; Fig. EV1C), referred to as the a2-loop. The a2-loop contains two highly acidic regions with sulfated tyrosines: the N-terminal region $_{659}$DDDEDSY*EIFE$_{669}$ and the C-terminal region $_{686}$EPEDEESDADY*DY*$_{698}$ (* denotes sulphation). The N-terminal region of the a2-loop was found to interact with the heparin-binding site of fXa, but the C-terminal region was not defined in the map of the prothrombinase complex.

Although the structure of the M17-prothrombinase complex advanced our understanding of prothrombinase assembly, it did little to address the fundamental questions regarding the cofactor function of fVa, namely: how is initial cleavage at Arg320 enforced; how is Arg271 presented to the active site of fXa after cleavage of Arg320; what conformational rearrangements ensure processivity (i.e., presentation of second cleavage site without dissociation of the intermediate); and how is high catalytic efficiency achieved? To address these issues, we solved the cryo-EM structures of M17-prothrombinase with the substrate prothrombin and with the intermediate meizothrombin, both to 3.1 Å resolution. The structures reveal a multi-step mechanism driven by allosteric conformational rearrangements to the substrate and its interaction with prothrombinase, providing a surprising and satisfying answer to the questions regarding the cofactor function of fVa. In this mechanism, the substrate is not a passive object acted upon by the enzyme, but an active participant in its own processing.

## Results

### Cryo-EM structure of the prothrombinase-prothrombin complex

The cryo-EM map exhibited features consistent with an overall resolution of 3.1 Å, with excellent coverage of all domains for all three proteins, signal corresponding to glycosylation, and most side chains resolved (Fig. 1). The prothrombinase component was essentially identical to our previous structure of the substrate-free complex (Ustok et al, 2026), with a root-mean square deviation (RMSD) of 0.64 Å for 1452 equivalent Cα atoms and a conserved fVa-fXa interface (Appendix Tables S1 and S2) that buries a total of 4973 Å². Prothrombin was found exclusively in the "closed" conformation (Pozzi et al, 2016; Pozzi et al, 2013), with the F1 fragment interacting with the active site of the SP domain, and no evidence in any of the classes of the open form where the linker

between the K1 and K2 domains is extended. Nevertheless, the Gla domain of prothrombin is roughly co-planar with the membrane-binding domains of prothrombinase, enabling PL interaction for the ternary complex with a small tilt forward for prothrombinase. The complex between prothrombin and prothrombinase is 'productive', with the 320-loop bound in the active site of fXa as a normal substrate (Fig. EV2A), and therefore represents the recognition complex of the initial cleavage event (note that the catalytic serine of fXa was mutated to alanine to prevent hydrolysis). The interface between prothrombin and fVa buries a total of 1780 Å² and involves two principal regions: the stretch between the A1 and A2 domains of fVa (313–320; the a1-loop) with the loop preceding Arg271 of prothrombin (253–266; the 271-loop); and the C-terminal region of the a2-loop of fVa with exosite I of prothrombin (Appendix Table S3). Surprisingly, the a2-loop accounts for 71% of the total buried surface area (BSA) between fVa and prothrombin, underscoring the importance of this contact for substrate recognition. Similarly, the fXa-prothrombin interface (total BSA of 1458 Å²) is dominated by the interaction of the substrate loop ($_{317}$IEGR-IV$_{322}$) with the active site of fXa (Appendix Tables S4 and S5), accounting for 77% of the total BSA. The docking of prothrombin to prothrombinase, therefore, involves loops interacting with loops or loops interacting with an ordered domain; there are no substantive interactions involving ordered domains of prothrombin with ordered domains of prothrombinase. This "light–touch" interaction is typical of a substrate, where stable complex formation would inhibit turnover. Also consistent with an efficient substrate-enzyme interaction is an interface dominated by electrostatics (Fig. 2A,B). This is exemplified by the basic a1-loop of fVa ($_{303}$KKTRNLKKI-TREQRRHMKR$_{321}$) interacting with the acidic 271-loop of prothrombin ($_{249}$EEAVEEETGDGLDEDSDRAIE$_{269}$) (Fig. 2C) and the acidic a2-loop of fVa (685–709) interacting with the basic exosite I of prothrombin (Fig. 2D).

### The a2-loop:prothrombin interaction

The interaction between the C-terminal region of the a2-loop and prothrombin was unanticipated, and even more surprising was its contribution of more than 70% to the total interaction interface between fVa and prothrombin. Although this region has been implicated in prothrombin processing (Kalafatis et al, 2003), the assumption was that it interacted with fXa (Shim et al, 2015; Toso and Camire, 2006). We were able to build the a2-loop onto prothrombin using the cryo-EM map on its own; however, due to the unexpected nature of this interaction and to test the accuracy of its placement, we assessed the interaction computationally and structurally. AlphaFold 3 (Abramson et al, 2024) was run using the sequences of Pre-2, the zymogen form of the SP domain of prothrombin, and a C-terminal a2-loop peptide ($_{686}$EPEDEESDA-DY*DY*QNRLAAALGIR$_{709}$; * denotes phosphorylation). The five outputs were identical from 690 to 709 (Fig. 3A), with a prediction

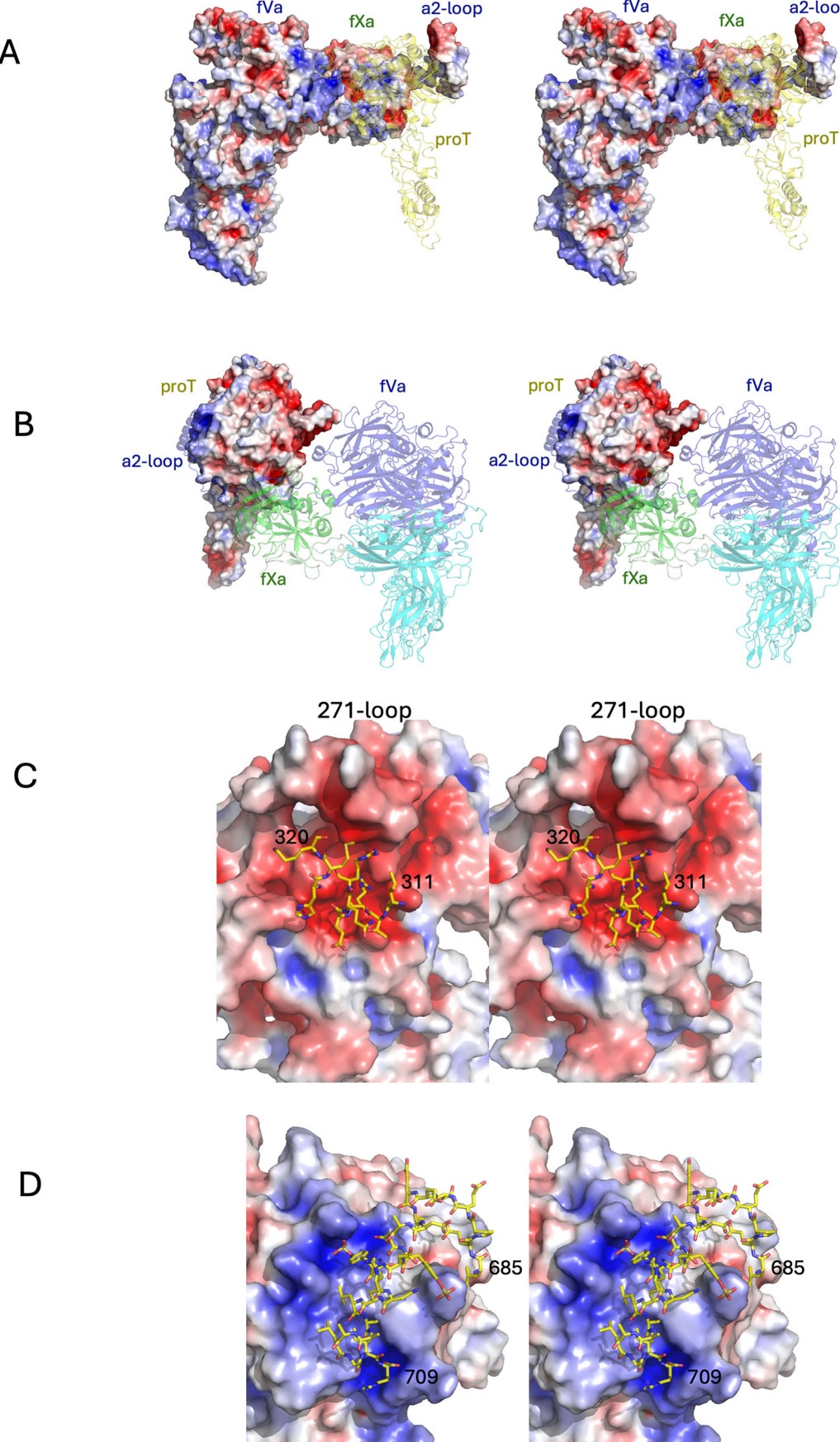

**Figure 2.  Stereo views of electrostatic surfaces of prothrombinase and prothrombin.**

(A) The surface of prothrombinase is colored blue for positive and red for negative electrostatic potential. Prothrombin is depicted as a semitransparent yellow cartoon. A large basic surface is presented to prothrombin by the body of fVa, and an acidic surface is presented with the a2-loop. (B) The complex is reoriented, with prothrombinase shown as semitransparent cartoons colored as in Fig. 1. The electrostatic surface of prothrombin is complementary to the surfaces presented by prothrombinase, with a large acidic patch facing the body of fVa and the basic exosite I of prothrombin interacting with the acidic a2-loop C-terminus. Domains are labeled for clarity. (C) Close up of the electrostatic surface of prothrombin (with the loop containing Arg271, the 271-loop, indicated) and the basic a1-loop of fVa (the loop connecting the A1 and A2 domains; residues 311-320) depicted as yellow sticks. (D) Close up of the electrostatic surface of prothrombin focussed on exosite I with the C-terminus of the a2-loop of fVa (residues 685–709) depicted as yellow sticks.

of high confidence (pIDTT >70; Appendix Fig. S1). We also obtained a 2.8 Å crystal structure of the peptide with Pre-2 (Fig. 3B). The AlphaFold result and the crystal structure corresponded well to the cryo-EM structure (Fig. 3C). The interaction between the C-terminal a2-loop and prothrombin is typical of exosite I interactions of thrombin with its ligands (Huntington, 2005), with significant binding site overlap and the utilization of core hydrophobic contacts. The most similar in structure is the sulfated N-terminal extension of the inhibitor heparin cofactor II (HCII; $_{55}$EEDDDY*LDLEKIFSEDDDY*$_{73}$) (Baglin et al, 2002). The acidic HCII loop forms a helix that runs in the opposite direction to place three hydrophobic side chains (Phe67, Leu63, and Leu61) into the same positions as Leu702, Leu706, and Ile708 of the a2-loop on a hydrophobic surface of exosite I, comprised of Leu380, Ile398 and Met400 (residues 65, 82, and 84 in chymotrypsin numbering; Fig. EV3). The importance of exosite I in prothrombin processing was established over 25 years ago with the observation that hirugen was able to reduce the rate of thrombin generation to levels obtained in the absence of fVa (Anderson et al, 2000). Our finding that the a2-loop binds to exosite I of prothrombin and constitutes the principal contact between fVa and prothrombin fully explains that observation.

## Cryo-EM structure of the prothrombinase-meizothrombin complex

We obtained a map of the prothrombinase-meizothrombin complex to 3.1 Å resolution, employing the same conditions used for the prothrombin complex. Two surprises were immediately evident—the F1 fragment of meizothrombin and the C-terminal region of the a2-loop of fVa were both absent from the map. No particle subset possessed either feature, nor did they appear as a weak signal captured using a low-resolution filter. Otherwise, the map covers all domains and is of similar quality and resolution as the map with prothrombin (Fig. 4). Importantly, the complex is productive, with the map providing clear evidence that the 271-loop is presented as a substrate in the active site of fXa (Fig. EV2B) in a manner similar to the 320-loop of prothrombin (Fig. EV2C). The prothrombinase component of the meizothrombin complex is again identical to substrate-free prothrombinase (RMSD of 0.8 Å for 1455 Cα atoms) and, other than loss of the C-terminus of the a2-loop, is also identical to the structure with prothrombin (RMSD 0.61 Å). However, the two domains observable for meizothrombin (K2 and SP) have shifted substantially and make entirely new contacts with prothrombinase. The contact between meizothrombin and fVa is limited in nature with a total BSA of 327 Å², involving only the 271-loop of meizothrombin (contacting residues 261–266) with the a1-loop of fVa (contacting residues 314–318;

Appendix Table S6). The contact between meizothrombin and fXa is more extensive, with a total BSA of 2187 Å², half of which involves the substrate loop ($_{268}$IEGR-TA$_{273}$) with the active site of fXa (Appendix Table S7). The other interactions between meizothrombin and fXa can be considered exosite interactions, and primarily consist of the remnant of the 320-loop of meizothrombin (310–320) with the 147-loop of fXa (145–150). The only contacts involving the SP domain of meizothrombin also engage the 147-loop of fXa (Appendix Table S8). Despite the large shift in position of meizothrombin relative to that of prothrombin, the interface still exhibits electrostatic complementarity (Fig. 5A,B).

## The prothrombin to meizothrombin conformational change

It is worth reiterating that the prothrombinase complex remains unaltered during prothrombin binding and processing (Appendix Fig. S2), and that only the substrate undergoes conformational rearrangements. Focusing on the SP domain, conversion of prothrombin to meizothrombin is associated with a 50° rotation and a 10 Å shift (Fig. 6A,B and Movies EV1 and EV2). This reorientation is triggered by the zymogen-to-protease conformational change (Wang et al, 1985) in the SP domain following insertion of the new N-terminus ($_{321}$IVEG$_{324}$; 16–19 in chymotrypsin numbering) into the core of the SP domain. Activation of meizothrombin results in dramatic alterations to the shape and properties of the active site region (Fig. 7A) and converts extensive interactions with K1 (Appendix Table S9) into severe clashes (Appendix Table S10), explaining the loss of contact and conversion to the open configuration, consistent with recent findings (Stojanovski et al, 2018). These active site rearrangements are common to all serine proteases upon activation, so loss of the active site interaction with F1 is easily rationalized. In contrast, although there are changes evident in exosite I (Fig. 7B), it is not clear why the interaction with the a2-loop is no longer observed. However, exosite I is known to be sensitive to the zymogen-to-protease conformational change and has been described as a pro-exosite due to the inability of prothrombin to bind to exosite I ligands of thrombin (Anderson et al, 2003; Kroh et al, 2007), such as thrombomodulin (Wu et al, 1994) and fibrinogen (Kaczmarek et al, 1987).

Of particular relevance for the repositioning of meizothrombin to present Arg271 for cleavage is the movement of the K2 domain to alter the properties of the surface facing fVa (Fig. 7C). The K2 domain shifts 12 Å relative to the SP domain (Fig. 7D; Movie EV3), resulting in a change in shape and electrostatics (Fig. 7C). In prothrombin, K2 interacts primarily with the C-terminal helix of the SP domain, centered on Lys572 (Lys240 in chymotrypsin

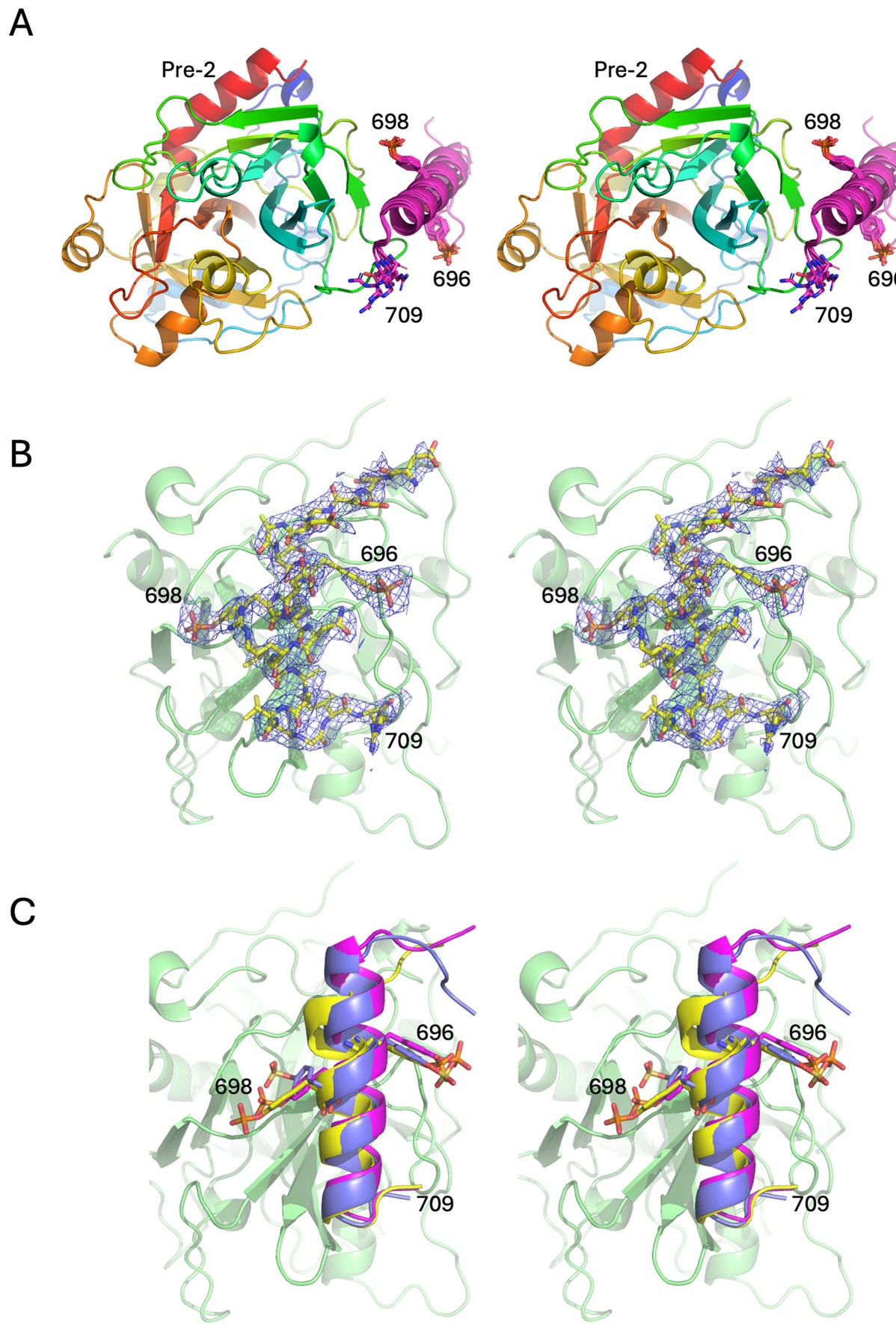

**Figure 3. Stereo views of the predicted and experimentally-derived interactions between prothrombin and the C-terminal portion of the a2-loop.**

(A) The AlphaFold models of the Pre-2 complex with an a2-loop peptide. Pre-2 is colored from N-to-C-terminus (blue-to-red; in the standard orientation) and the five a2-loop predictions are in magenta, with the C-terminal Arg709 and the two phosphorylated Tyr residues depicted as sticks. (B) Electron density (blue mesh) surrounding the a2-loop peptide (yellow sticks) from the crystal structure of its complex with Pre-2 (green cartoon). (C) Cartoon depictions of the a2-loop regions from the AlphaFold solution (magenta), the crystal structure (yellow), and the cryo-EM structure of the prothrombinase-prothrombin complex (blue) demonstrate the overall agreement (oriented as in (B) with prothrombin in green).

numbering) (Appendix Table S11), but shifts in meizothrombin to interact primarily with the 90-loop centered on Arg93 (409 in prothrombin numbering; Appendix Table S12). Changes in the surface properties of exosite II upon activation of prothrombin to meizothrombin are evident (Fig. 7D), and some clashes with K2 are engendered (Appendix Table S13), which together appear to drive the repositioning of K2. One other conformational change of note in conversion from prothrombin to meizothrombin is the formation of a helix in the loop N-terminal to Arg320 (Fig. EV4). This new helix associates with the SP domain of meizothrombin and interacts with the 147-loop of fXa, constituting an important exosite interaction (Appendix Table S7).

## Discussion

The complexity of the blood coagulation cascade and its multiple feedback loops exists to regulate thrombin generation (Davie and Ratnoff, 1964; Macfarlane, 1964). Small perturbations can result in clinical manifestations of bleeding or thrombosis, and severe dysregulation, as in the case of snake venom toxins, can cause systemic thrombosis and hemorrhage (Kini, 2005). Key to limiting thrombin generation to sites of vascular damage is a dependency on activated PL surfaces, not usually in contact with the blood, for assembly of the prothrombinase complex. Prothrombin activation is unusual amongst the many cleavage-activation events of the blood coagulation cascade because it requires the severing of two remote bonds. How prothrombinase processes prothrombin at two sites in a rapid and processive manner is a mechanistic mystery that we have largely solved by obtaining cryo-EM structures of prothrombinase with its substrate and intermediate.

One of the longstanding questions regarding prothrombin processing relates to the order of cleavage. In the absence of the cofactor, fXa cleaves prothrombin very slowly at both sites, with Arg271 kinetically favored over Arg320 (Krishnaswamy, 2013). Other than a small kinetic preference for the 271 site, there is nothing sequential about the two cleavage events (i.e., cleavage of one site does not influence cleavage at the second site). When fVa is added to the reaction, evidence for initial cleavage at Arg320 begins to appear, with the fraction processed via meizothrombin increasing with fVa concentration (Boskovic et al, 1990). PL surfaces are required for most of the processing to occur via initial cleavage at Arg320, although there is still evidence of initial cleavage at Arg271 in vitro. We recently showed that it was possible to recapitulate the effect of PL addition simply by mutating fXa to improve its affinity for fVa (Ustok and Huntington, 2026). The PL surface, therefore, is unlikely to exert any influence on prothrombin processing beyond improving the association of the two components of prothrombinase. Therefore, cryo-EM structures obtained in the absence of PL using M17 fXa should have the features of PL-

bound prothrombinase required for prothrombin binding and processing.

Cryo-EM captures a snapshot of a biological system, with different classes of complexes representing the distribution of states at the time of vitrification. It is interesting to note, that although we observed several possible classes for the ternary prothrombin complex, they reflected only a small degree of rotational freedom pivoting on Arg320 in the active site cleft of fXa (Fig. EV5), with no observed particle class presenting Arg271 for cleavage. This supports exclusive processing of prothrombin at Arg320 by assembled prothrombinase (Ustok et al, 2024), and suggests that evidence of initial cleavage at Arg271 in vitro is likely an artifact due to a fraction of free fXa. The apparent rotational freedom of prothrombin bound to prothrombinase also has important functional implications. The light–touch interaction between prothrombin and prothrombinase is unlike a normal stable protein–protein complex, with prothrombin appearing to be trapped in an electrostatic cage that enforces the presentation of the Arg320 cleavage site. Prothrombin is free to move within the cage so long as the orientation is roughly the same and Arg320 is engaged in the active site of fXa. Complementary electrostatics are known to accelerate substrate association (Schreiber et al, 2009; Waldner et al, 2018) and help explain the cofactor effect of fVa. The non-directional aspect of electrostatic contacts also allows prothrombin to undergo profound conformational rearrangements after the first cleavage event to present Arg271 without dissociating from prothrombinase. A substrate binding mode where rigid domains interact with high affinity would inhibit or slow the conformational rearrangement, and would require dissociation and reassociation to present the second cleavage site. Such a scenario would slow turnover and risk the loss of processivity. A loosely held substrate is also consistent with the rapid product dissociation necessary to achieve high turnover.

Now that we have determined the structures of prothrombinase in all three states, substrate-free, substrate-bound and intermediate-bound, we are able to piece together the mechanistic detail of the processive processing of prothrombin, previously described as "ratchetting" (Bianchini et al, 2005). Under physiological conditions, the components of prothrombinase will assemble on an activated cell, such as a platelet, that expresses phosphatidylserine on its surface. The two membrane-binding C domains of fVa and the Gla domain of fXa align in a linear manner. Prothrombinase is therefore able to pivot forward or backward to some degree, and need not bind to the membrane surface in a strictly perpendicular orientation. Fluorescence resonance energy transfer studies produced distances consistent with an average prothrombinase orientation roughly perpendicular to the membrane surface (Husten et al, 1987; Isaacs et al, 1986). However, the situation changes when prothrombin binds to prothrombinase. The closed conformation of prothrombin predominates in solution, and we

# Side view

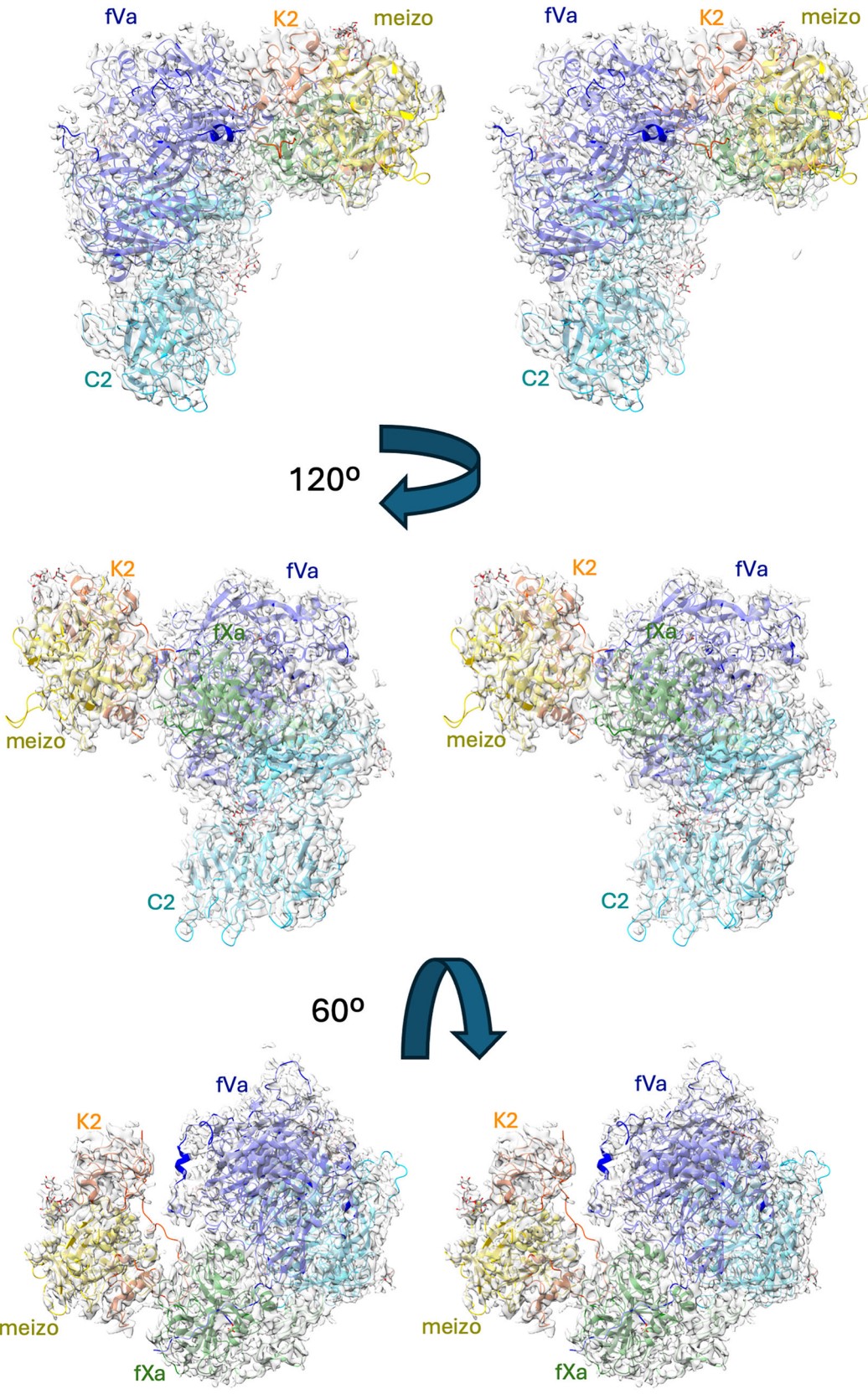

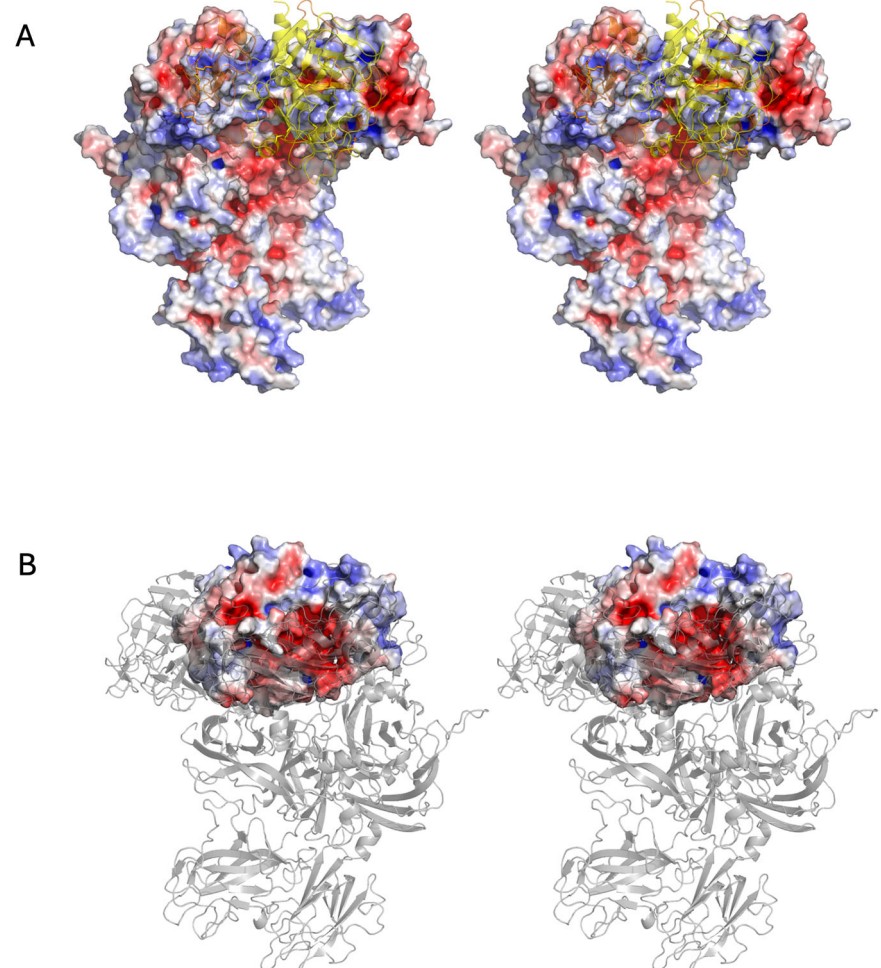

**Figure 4.   Stereo views of the structure of the prothrombinase-meizothrombin complex with cryo-EM map.**

Three views of the final coordinates of fVa, fXa and meizothrombin (meizo) with the surrounding map are shown, colored and oriented as in Fig. 1. The light chain of meizothrombin, including the K2 domain, is colored orange to distinguish it from the SP domain, which is in yellow. There is no map corresponding to the F1 region of meizothrombin or to the C-terminal portion of the a2-loop. Molecules and certain domains are labeled for clarity, and the figure was made using ChimeraX.

**Figure 5.   Stereo views of electrostatic surfaces of prothrombinase and meizothrombin.**

**(A)** The surface of prothrombinase is colored blue for positive and red for negative electrostatic potential. Meizothrombin is depicted as a semitransparent cartoon, colored yellow for the SP domain and orange for the light chain, including the K2 domain. The interaction surface on prothrombinase is overwhelmingly basic. **(B)** The complex is reoriented to look through the semitransparent cartoon representation of prothrombinase (gray) at the acidic electrostatic surface of meizothrombin.

only observe the closed configuration in our cryo-EM map, with no evidence of a smaller volume corresponding to prothrombin missing F1 in any of the particle subsets. Due to the compact state of prothrombin, prothrombinase must tilt forward by about 20° to engage a prothrombin molecule bound to the same membrane surface (Appendix Fig. S3).

The first state of the catalytic cycle (Fig. 8) is therefore represented by the prothrombinase-prothrombin cryo-EM structure bound to a PL surface, with prothrombinase leaning forward at an angle of about 64° relative to the PL plane (Fig. 8A). Cleavage of the Arg320-Ile321 bond then initiates a series of changes that begin

with the expulsion of the new N-terminus from the active site of fXa. Collapse of the tetrahedral transition state to generate the new C-terminal carboxylate on Arg320 and a protonated N-terminus on Ile321 engenders an extreme steric clash that necessitates the movement of one or both of the new ends. We propose that the new N-terminus ($_{321}$IV$_{324}$) will be expelled from the active site of fXa and that the new C-terminus ($_{317}$IEGR$_{320}$) will remain bound. This is due to the extensive nature of the contacts on the non-prime side (P4-P1; nomenclature of Schechter and Berger (Schechter and Berger, 1967)), especially Arg320 in the deep S1 pocket and Ile317 in the S4 pocket, relative to the weak interactions on the prime side.

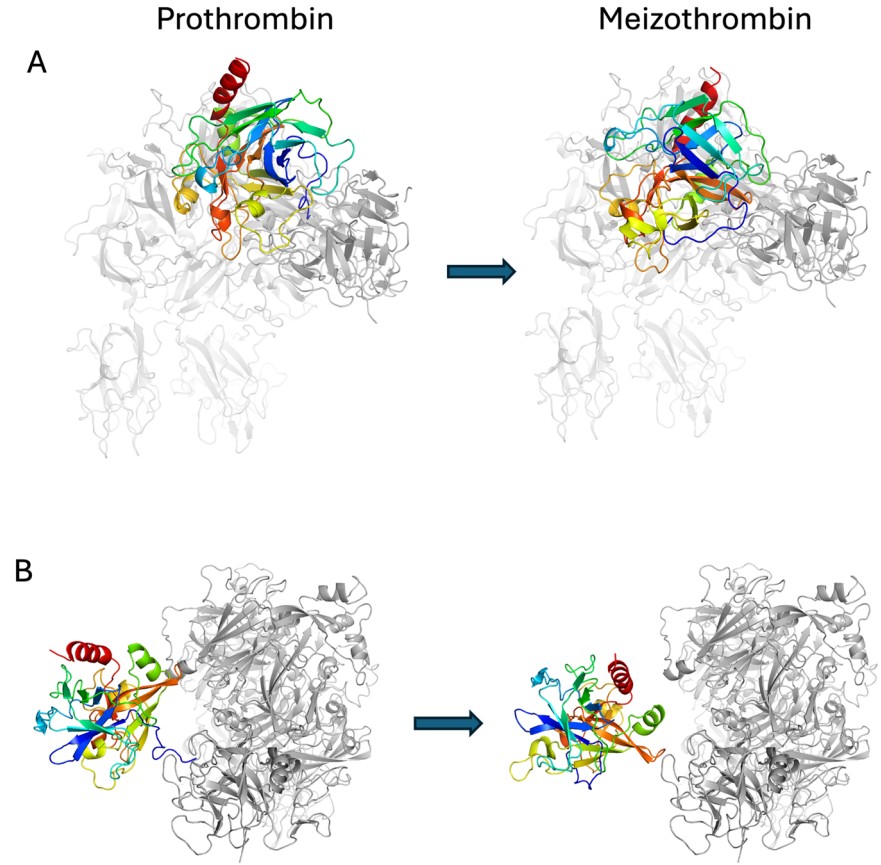

Prothrombin → Meizothrombin

**Figure 6. Movement and rotation of the SP domain of prothrombin upon conversion to meizothrombin.**

The SP domain of prothrombin (left) and meizothrombin (right) are depicted in cartoon representation, colored from N-to-C-terminus (blue-to-red) to illustrate the magnitude of positional and rotational shift that occurs after cleavage of Arg320. The side view (**A**) and top view (**B**) are shown, with prothrombinase colored gray. Animations of the morph in each orientation is provided in Movies EV1 and EV2.

Once the new N-terminus is free, it becomes a tethered ligand and rapidly finds its way to the center of the SP domain of prothrombin. The zymogen-to-protease conformational change then occurs to the SP domain (Wang et al, 1985), which, due to the allosteric linkage between the active site, exosite I and exosite II of prothrombin/thrombin (Anderson and Bock, 2003; Anderson et al, 2003; Billur et al, 2017; Fredenburgh et al, 1997; Kroh et al, 2007; Wu et al, 1994), all contacts involving the SP domain are dramatically altered. The dissociation of F1 from the SP domain and the shifting of the K2 domain on exosite II of the SP domain are likely to be simultaneous, and coincide with the conformational change in the SP domain. We propose that the K2 domain shift triggers the 50° rotation of the SP domain relative to prothrombinase, and that this rotation extracts the C-terminus of the light chain of meizothrombin ($_{317}$IEGR$_{320}$) from the active site of fXa. The extracted C-terminus then forms a helix to mediate a new exosite contact with the 147-loop of fXa. The rotation of the K2/SP unit also, importantly, places Arg271 into the active site of fXa (Movie EV4).

The release of F1 from the SP domain of meizothrombin appears to be necessary to allow the rotation of the SP domain while maintaining Gla domain contact with the PL (Appendix Fig. S4). Once dissociated from the SP domain, F1 will also need to shift towards the C2 domain of fVa in order to remain bound to the PL surface, due to the limited length of the linker between K1 and K2. Elongation of the linker and movement of F1 towards fVa will allow prothrombinase to tilt backward to obtain a more perpendicular orientation with respect to the PL surface. This PL-bound form of the meizothrombin-prothrombinase complex represents State 2 in the catalytic cycle (Fig. 8B).

It is not clear if the conformational change to exosite I causes the release of the a2-loop or if the rotation of the SP domain is responsible for breaking the contact. However, it is abundantly clear that meizothrombin is no longer bound to the a2-loop in our structure. We propose that its dissociation plays a role in potentiating high turnover. The first obvious effect of losing the a2-loop contact is that meizothrombin is held less firmly in place and will dissociate faster after cleavage of Arg271. The second and less obvious effect is that the release of the C-terminus of the a2-loop allows it to bind to a prothrombin molecule before dissociation of fully processed thrombin. This pre-loading of prothrombin from either the PL-bound or free pools would increase the rate of turnover and constitutes State 3 in the catalytic cycle (Fig. 8C).

Prothrombin                                        Meizothrombin

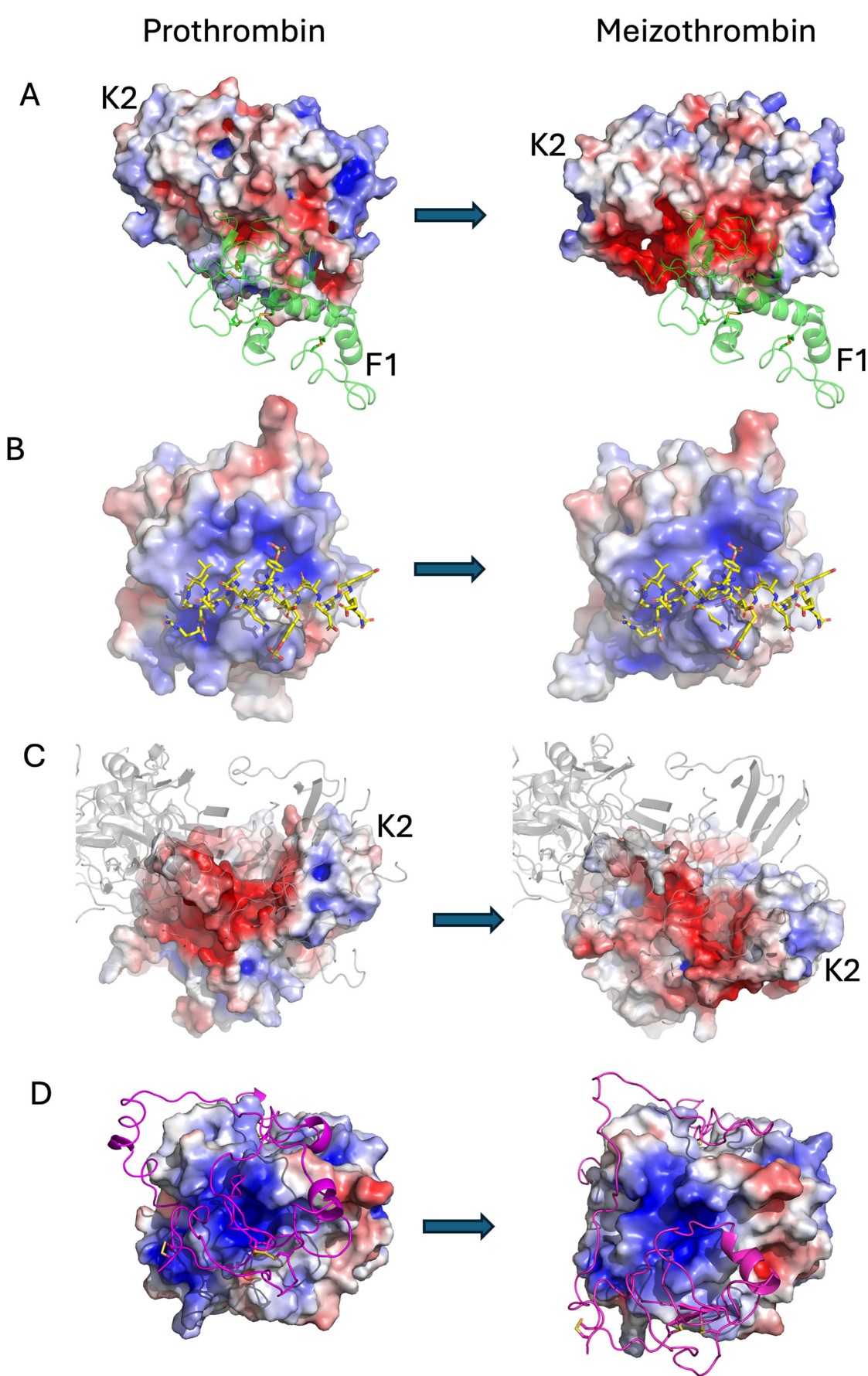

**Figure 7.  Surface electrostatic representations of the SP and K2 domains of prothrombin (left) and meizothrombin (right) with interaction partners.**

(A) The F1 region of prothrombin (semitransparent green cartoon; labeled) binds in the active site of prothrombin (SP domain in standard orientation), but is expelled upon conversion to meizothrombin (F1 is still depicted to illustrate changes to the interface). Major changes to the shape and electrostatic properties are evident, as is the movement of the K2 domain (labeled). (B) The surface of exosite I is depicted with the a2-loop in yellow sticks. Changes to the properties of exosite I are evident upon conversion to meizothrombin (the a2-loop is still depicted, although not observed in the meizothrombin structure). (C) The surface of the SP and K2 domains of prothrombin and meizothrombin presented to fVa (gray cartoon; from its complex with prothrombin), with the SP domains fixed, illustrates the dramatic shift in shape and properties upon cleavage of Arg320. The K2 domains are indicated. (D) The SP domains are oriented identically, with electrostatic surfaces shown. The K2 domain and the loop connecting it to the SP domain are depicted as a magenta cartoon. Exosite II is facing, and the movement of K2 along the SP domain is evident.

Meizothrombin is thus poised to dissociate as thrombin once cleavage at Arg271 occurs. As for the first cleavage event, the principal contacts in the active site of fXa are on the non-primed side. The prime side, now constituting the light chain of thrombin, will be expelled once cleavage has occurred. F1.2 is now only attached to thrombin by non-covalent interactions between K2 and exosite II, and in the absence of tethering, this interaction is very weak with a dissociation constant of 3.7 μM (Bradford and Krishnaswamy, 2016). Thrombin is therefore free to dissociate, leaving the C-terminus of F1.2 still in the active site of fXa, constituting State 4 of the catalytic cycle (Fig. 8D). F1.2 will then diffuse away as the precaptured prothrombin docks to begin another round of the catalytic cycle.

An unexpected implication of this work relates to the subject of thrombin allostery. Thrombin has long been considered an allosteric enzyme, with cross-talk between the active site, the Na$^+$ binding site, exosite I and exosite II (Fredenburgh and Weitz, 2025). However, there is no evidence that allostery is actually important for any of the multiple activities of thrombin, including the most radical switch in activity from pro- to anti-coagulant upon thrombomodulin binding. This and other activity changes in thrombin rely on exosite competition and not on allosteric switching (Huntington, 2005). What is abundantly clear from the prothrombinase mechanism elucidated here is that allostery plays an essential role in prothrombin processing. The apparent allosteric connections between the active site and exosites on thrombin may simply be the remnants of the allostery required for sequential and rapid processing of its precursor prothrombin, and may be of no further functional relevance.

Finally, it should be noted that two cryo-EM structures purporting to be of the prothrombinase-prothrombin complex have been deposited (Ruben et al, 2022; Stojanovski et al, 2024). We have previously detailed the shortcomings of 7TPP (Huntington et al, 2025), which is a low-resolution structure (likely 5.5 Å [PubPeer]) from grids made in the absence of PL. The deposition 7TPQ, originally of substrate-free prothrombinase formed on nanodiscs, has been replaced with 9CTH, which claims to be of the prothrombin-prothrombinase complex on nanodiscs at 6.5 Å resolution (Stojanovski et al, 2024). However, careful analysis has demonstrated that the prothrombin component is entirely missing from the map, and likely represents a case of Einstein-from-noise (Henderson, 2013) (PubPeer). We conclude that the structures presented in this manuscript are the first of sufficient quality and resolution to decipher the complex and surprising mechanism by which prothrombin actively participates in its own processing, a mechanism we term "substrate allostery".

# Methods

### Reagents and tools table

| Reagent/resource | Reference or source | Identifier or catalog number |
|---|---|---|
| **Experimental models** | | |
| HEK-EBNA (Human) | Life Technologies | 300264 |
| BHK-M (Mesocricetus auratus, Golden hamster) | ATCC, PTA-4506 | RRID:CVCL-4U20 |
| BL21 STAR DE3 plysS (E. coli) | Invitrogen | Cat# C606010 |
| Max Efficiency DH5α (E. coli) | Invitrogen | Cat# 18258012 |
| **Recombinant DNA** | | |
| pSV2Neo | ATCC | Cat# 37149 |
| pED-rfV DT | Toso and Camire, 2004 | |
| pCEP4-hII S195A | Ustok and Huntington 2022, 2026 | |
| pET23-Pre-2 S195A | Adams and Huntington, 2016; Johnson et al, 2005 | |
| pET23-M17(EGF2-SP) S195A | This study | |
| **Antibodies** | | |
| **Oligonucleotides and other sequence-based reagents** | | |
| Custom Primers | Sigma-Aldrich | Appendix Table S14 |
| **Chemicals, enzymes and other reagents** | | |
| Gibson Assembly Cloning Kit | New England Biolabs | Cat# E5510S |
| Q5 Hot Start High Fidelity 2X Master Mix | New England Biolabs | Cat# M0494S |
| QIAprep Spin, miniprep kit | QIAGEN | Cat# 27104 |
| QIAquick PCR purification kit | QIAGEN | Cat# 28104 |
| QIAquick gel extraction kit | QIAGEN | Cat# 28704 |
| CD CHO medium | Gibco | Cat# 10743029 |
| L-Glutamine | Sigma-Aldrich | Cat# G7513-100ml |
| Anti-anti (100x) | Gibco | Cat# 15240-062 |
| Vitamin K1 | Sigma | Cat# V3501-1G |
| Hygromycin B | Gibco | Cat# 10687010 |
| DMEM/F12 | Gibco | Cat# 2104-025 |
| ITS-A (100x) | Gibco | Cat# 51300044 |

| Reagent/resource | Reference or source | Identifier or catalog number |
|---|---|---|
| Albumax-I Lipid-rich BSA | Gibco | Cat# 11020062 |
| G418-Disulphate | Formedium | Cat# G4185 |
| RVV-X | Latoxan | Cat# L8134-50U |
| Thrombin from human plasma | Roche | Cat# 10602400001 |
| Benzamidine-HCl | Sigma-Aldrich | Cat# B6506-100G |
| AEBSF | Thermo Scientific | Cat# 78431 |
| cOmplete Mini, EDTA-free | Roche | REF 11836170001 |
| IPTG, Isopropyl β-D-thiogalactoside | Sigma-Aldrich | Cat# I6758-10G |
| SP Sepharose, FF resin | Cytiva | Cat# 17072910 |
| HiTrap Q HP column | Cytiva | Cat# 17115301 |
| HiTrap Heparin HP column | Cytiva | Cat# 17040701 |
| Superdex 200 pg 16/600 | Cytiva | Cat# 28989335 |
| Quantifoil (Au) R1/1 on Au, 300 mesh | Quantifoil Micro Tools | Cat# N1-A21nAu30-50 |
| Ecarin | Enzyme Research Laboratories | Cat# 116-01 |
| Bolt™ Bis-Tris Plus Mini Protein Gels, 4-12%, 1.0 mm, WedgeWell™ format | Thermo Fisher Scientific | Cat# NW04120BOX |
| InstantBlue, Coomassie Protein Stain | Abcam | Cat# ab119211 |
| MES-SDS running Buffer 20X | Formedium | Cat# MES-SDS1000 |
| Index Screen | Hampton Research | Cat# HR2-144 |
| **Software** | | |
| AlphaFold 3 server | https://alphafoldserver.com/welcome | |
| AIMLESS (CCP4: supported program) | https://www.ccp4.ac.uk | |
| CCP4-suite | https://www.ccp4.ac.uk | |
| ChimeraX | https://www.cgl.ucsf.edu/chimera/ | |
| CryoSparc v4.7 | https://cryosparc.com/ | |
| Coot | https://www2.mrc-lmb.cam.ac.uk/personal/pemsley/coot/ | |
| MolProbity | http://molprobity.biochem.duke.edu | |
| Phaser crystallographic software | https://www.ccp4.ac.uk | |
| Phenix | https://phenix-online.org | |
| Pymol (Schrodinger) | https://www.pymol.org | |
| Refmac (CCP4; supported program) | https://www.ccp4.ac.uk | |
| **Other** | | |

| Reagent/resource | Reference or source | Identifier or catalog number |
|---|---|---|
| DNA-Sequencing service | GENEWIZ from Azenta Life Sciences | |
| Vitrobot Mark IV | Thermo Fisher Scientific | |
| Pelco easiGlow, Glow discharge system | Pelco | |
| Titan Krios | Thermo Fisher Scientific | |
| AKTA Purifier | GE Life Sciences | |
| Mosquito LCP | TTP Labtech | |
| Rock Imager 182 | Formulatrix | |
| Vivaflow® 200 TFF Cassettes,10 kDa | Sartorius | Cat# VF20PO |
| Amicon 10 K filter units | Millipore Sigma | Cat# UFC901008 |

## Recombinant protein expression and purification

Human B-domainless fV was expressed, purified and activated as previously (Toso and Camire, 2004). The minimal M17 human fXa construct containing only the EGF2 and SP domains was used in place of the full-length protein because the Gla and EGF1 domains contribute little to fVa binding (Ustok and Huntington, 2026) and were not well defined in the map of our previous cryo-EM structure (Ustok et al, 2026). The S195A variant of M17 fX (assembled as in Appendix Tables S14 and S15) was produced in E. coli as inclusion bodies, refolded, purified, and activated as previously described (Ustok et al, 2024). Human prothrombin (S195A) was prepared as described previously (Ustok and Huntington, 2022). Meizothrombin was generated from S195A prothrombin as previously (Tans et al, 1994). The purity and integrity of meizothrombin was confirmed by SDS-PAGE.

## Cryo-EM grid preparation, data collection, processing, and refinement

Grid preparation and data collection were conducted essentially as previously (Ustok et al, 2026). Briefly, fVa was mixed with truncated S195A fXa (M17 variant) and S195A prothrombin or meizothrombin at a 1:6:2 molar ratio, with final concentrations of 650 nM, 3.9 and 1.3 μM, respectively. All proteins were in 20 mM HEPES pH 7.5, 150 mM NaCl, and 5 mM $CaCl_2$. Gold grids (Quantifoil (AU) R1/1 on Au 300 mesh) were glow-discharged in residual air at 25 A for 60 s using a Pelco EasiGlow, and 3 μl of protein complex solution was applied. Following a 10 s wait time, grids were blotted for 1 s using −7 (prothrombin) or 0 (meizothrombin) blot force in a 100% humidity chamber at 4 ℃, and then flash plunged into liquid ethane (−180 ℃) using a Vitrobot Mark IV. The prothrombin complex was imaged using a Titan Krios electron microscope with a K3 detector (Gatan) at the Cambridge Nanoscience Centre. Data were collected at 130,000 magnification (pixel size 0.829 Å) at a dose of 50 e−/Å$^2$ and defocus values between −1.8 and −0.6 μm. Grids of the meizothrombin complex were imaged using a Titan Krios with a Falcon4i counting camera at the University of Cambridge, Department of Biochemistry

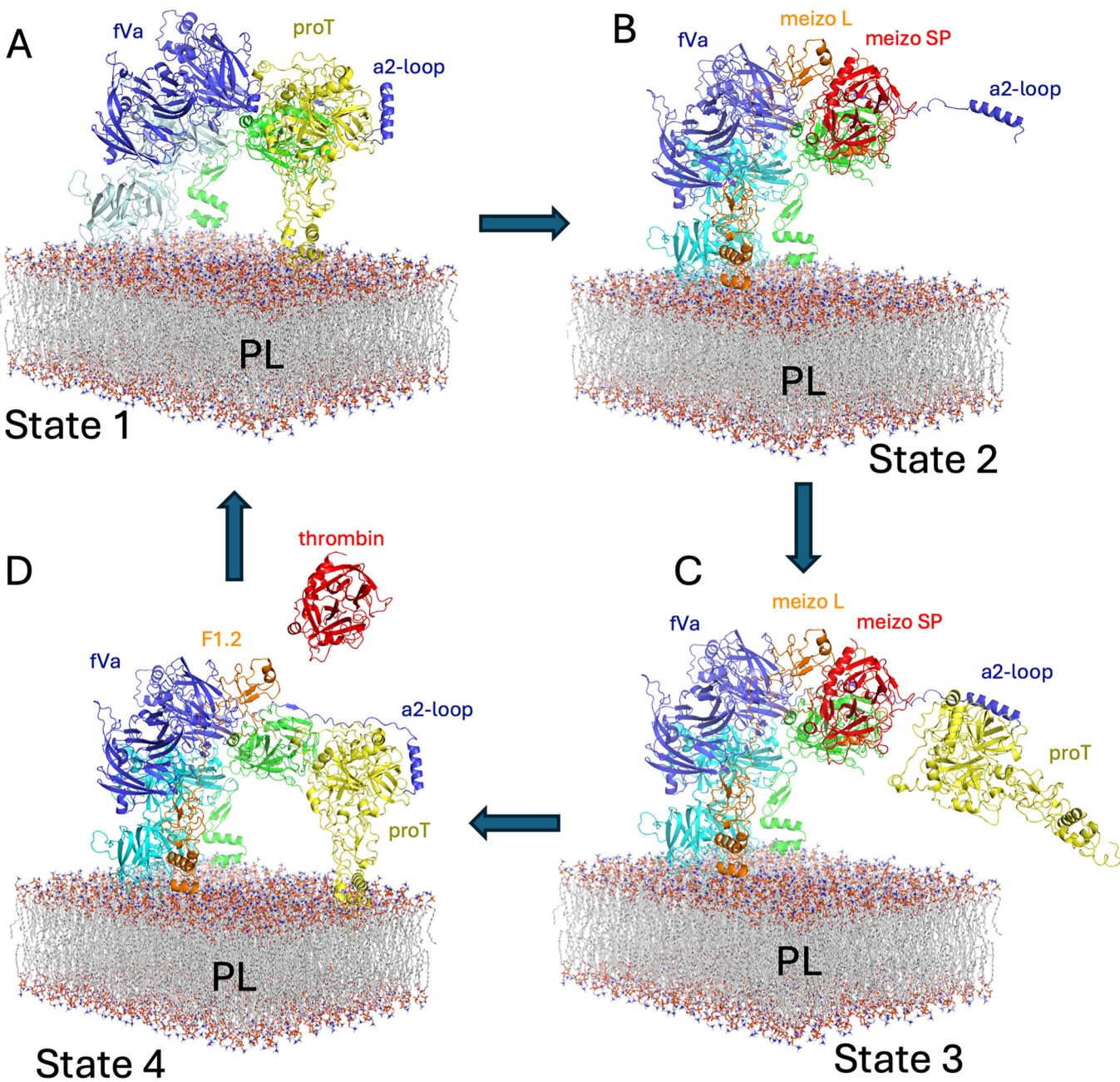

**Figure 8. The four states of the catalytic cycle of prothrombin processing by prothrombinase.**

The prothrombinase complex bound to a phospholipid (PL) bilayer is depicted, as in Fig. 1, with the Gla and EGF1 domains of fXa added. (**A**) Prothrombin (proT; yellow) bound to prothrombinase in a productive complex, with all three proteins binding the PL surface (State 1). Prothrombinase is oriented at an angle of about 64° relative to the plane of the PL membrane. (**B**) State 2 following cleavage at Arg320 to form the meizothrombin intermediate (meizo; orange for F1.2 and red for active SP domain). The Gla and EGF1 domains of the light chain of meizothrombin (meizo L) have moved to accommodate the rotation of the SP domain (meizo SP), and the C-terminal portion of the a2-loop has disengaged. Prothrombinase relaxes back to a near-perpendicular orientation with respect to the PL surface. (**C**) We hypothesize that before or during cleavage of Arg271 that the a2-loop will engage another prothrombin molecule from either the PL-bound or free (depicted) pool (State 3). The Kd of prothrombin for an activated PL surface is similar to its concentration in blood, so both pools should be populated (Smith et al, 2013). (**D**) Cleavage of Arg271 expels the product thrombin (red; State 4), and prothrombin can now move into the embrace of prothrombinase, attracted by long-range electrostatics, and the remnant F1.2 is free to diffuse away, returning to State 1 and completing the cycle.

Cryo-EM facility. Data were collected at 165,000 magnification (pixel size 0.729 Å) at a dose of 50.3 e−/Å² and defocus values between −1.8 and 0.8 μm. All data were collected on a tilted stage (−25°) to improve the particle orientation distribution.

All data were processed in cryoSPARC (Punjani et al, 2020). Patch motion correction and patch CTF were first run using default parameters; exposures with an estimated CTF resolution fit higher than 4.6 Å or a defocus tilt angle larger than 30 deg were discarded.

For the prothrombin complex, blob picker was used to pick and extract 7,138,515 particles that were extracted in 320 × 320 pixel boxes. Several rounds of ab initio reconstructions, heterogeneous refinements and 3D classifications were used to filter out bad particles and generate a 'consensus' reconstruction by homogeneous refinement based on a subset of 440,004 particles representing the intact fVa-fXa-prothrombin complex. These particles were re-extracted in 384 × 384 pixel boxes. A single round of masked 3D classification was run to sort particles depending on the orientation of prothrombin relative to the prothrombinase complex. The mask was generated from the "consensus" reconstruction and encompassed prothrombin as well as small parts of fXa and fVa using the following custom parameters: lowpass filter of 12 Å, threshold of 0.02, and soft padding width of 18 pixels. Five distinct classes were obtained (Fig. EV5). The best defined map was further refined using non-uniform refinement to a final resolution of 3.1 Å using 76,039 particles (Appendix Fig. S5A). The "consensus" subset was also used to obtain an improved map for prothrombin by masked local refinement. The mask described above was used again, along with a custom parameter of rotation search extent of 20 degrees, to generate a 3.4 Å map. A composite map was constructed using the original map focused on prothrombinase and the one focused on prothrombin. Briefly, most of the map corresponding to prothrombin was removed from the best overall map using ChimeraX (Pettersen et al, 2021) and sharpened in Phenix (Afonine et al, 2018). Similarly, the map corresponding to prothrombinase was removed from the prothrombin-focused map and then sharpened. A composite map was then made in Phenix. The resulting composite map preserved the best features of each and was used for model building. For the meizothrombin complex, a final set of 25,376 particles was similarly re-extracted to generate a final map of 3.1 Å resolution (Appendix Fig. S5B). A similar issue of rotational freedom of meizothrombin was observed in the map (Fig. EV6), and a composite map was made for model building, as before.

The M17-prothrombinase structure (9I2H) was fit into the map of the prothrombin complex in ChimeraX (Pettersen et al, 2021). The AlphaFold 3 (Abramson et al, 2024) model of prothrombin (AF-P00734-F1-v6) was chosen as a starting model for prothrombin, due to the lack of any high-quality crystal structure and the fact that it fit well into the cryo-EM map. The first 43 residues (signal sequence and propeptide) were removed, and the mature protein was renumbered. Gamma carboxylation was not performed on the coordinates because the map was of insufficient quality in that region to place side chains. The model of prothrombin was fit into the map using ChimeraX. The F1 of prothrombin and the two C domains of fVa were subjected to rigid body refinement in Coot (Emsley and Cowtan, 2004) and underwent small shifts relative to the starting position. Similarly, 9I2H was placed into the map of the meizothrombin complex in ChimeraX. It was immediately clear that the volume corresponding to meizothrombin did not contain the F1 region, and fit well to the crystal structure of desF1 meizothrombin 3E6P (Papaconstantinou et al, 2008), which was placed in the map using ChimeraX. Rigid body refinement was conducted in Phenix (Afonine et al, 2018) for each chain of meizothrombin and for the two C domains of fVa. Model building was conducted using Coot and refinement with Phenix. Data collection, processing and refinement statistics are provided in Tables EV1 and EV2.

## The Pre-2-a2-peptide crystal structure

An N-terminally acetylated peptide corresponding to the C-terminal region of the a2-loop (EPEDEESDADY\*DY\*QNRLAAALGIR; \* indicates phosphorylation; Biomatik) was dissolved in 10 mM Tris pH 8.0 to a final concentration of 10 mM. A fivefold molar excess of peptide was added to a solution containing 10 mg/ml Pre-2 to form a complex prior to crystallization. Crystallization was performed by sitting drop vapor diffusion in 96-well MRC crystallization plates (SWISSCI) using the Mosquito LCP (TTP Labtech) to dispense 100 nl of protein and 100 nl of reservoir solution. Crystallization was monitored using a Rock Imager 182 (Formulatrix). Crystals of various morphologies were obtained in several conditions and were cryoprotected with 20% glycerol, flash-cooled in liquid nitrogen and sent to Diamond Light Source (Oxford, UK) beamline I03 for data collection. The best dataset was obtained from a crystal grown in Index screen (Hampton Research; 0.1 M Tris, pH 8.5, 0.2 M NaCl, 25% PEG3350). Data processing were conducted using AIMLESS (Evans and Murshudov, 2013) and molecular replacement was conducted in Phaser (McCoy et al, 2007) with an AlphaFold 3 (Abramson et al, 2024) model of the Pre-2-peptide complex. Model building and refinement were conducted in Coot (Emsley and Cowtan, 2004) and Refmac (Murshudov et al, 2011), and structure validation was conducted with MolProbity (Williams et al, 2018). Processing and refinement statistics are given in Table EV3.

## Data availability

Maps and coordinates of the prothrombinase structures with prothrombin and meizothrombin are available from the Protein Data Bank (https://www.rcsb.org/) with accession numbers 9TLE and 9TLG, and the Electron Microscopy Databank (https://www.ebi.ac.uk/emdb/) with accession numbers 56052 and 56054. Coordinates and structure factors for the crystal structure of Pre-2 with the a2-loop peptide are available from the Protein Data Bank with accession number 9T00.

The source data of this paper are collected in the following database record: biostudies:S-SCDT-10_1038-S44318-026-00782-4.

## Peer review information

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

## Acknowledgements

This work was part-funded by British Heart Foundation Program (RG/16/9/32391) and Project (PG/24/11721) grants to JAH, and Cancer Research UK Discovery Award (DRCNPG-Jun24/100002) and UK Medical Research Council (UKRI1443) Program Funding to AJW. We thank Dima Chirgadze, Giulia Paris and Lee Cooper at the Department of Biochemistry Cryo-EM Facility, University of Cambridge and Sigurdur Thorkelsson, Pablo Castro Hartmann and Scott Gardner at the Cambridge Nanoscience Cryo-EM facility for their expert assistance.

## Author contributions

**Fatma Işık Üstok**: Conceptualization; Formal analysis; Investigation; Methodology; Project administration. **Alexandre Faille**: Data curation; Formal analysis; Methodology. **Alan J Warren**: Supervision; Funding acquisition. **James A Huntington**: Conceptualization; Formal analysis; Supervision; Funding acquisition; Validation; Investigation; Visualization; Writing—original draft; Project administration; Writing—review and editing.

Source data underlying figure panels in this paper may have individual authorship assigned. Where available, figure panel/source data authorship is listed in the following database record: biostudies:S-SCDT-10_1038-S44318-026-00782-4.

## Disclosure and competing interests statement

The authors declare no competing interests.

# Expanded View Figures

**Figure EV1.  Schematics of prothrombin processing and sequence of the a2-loop.**

(**A**) Schematics of prothrombin domain organization in open and closed states. The N-terminal gamma carboxyglutamic acid (Gla) domain and the first kringle domain (K1) are associated and are followed by a 26-residue linker and the K2 and serine protease (SP) domains that are also tightly associated. The closed form, where K1 interacts with the active site of the SP domain, predominates in solution. (**B**) Prothrombin processing pathways are shown with domains labeled as in (**A**). fXa, on its own will cleave first at Arg271 (left) to form prethrombin-2 (Pre-2) and F1.2, and prothrombinase cleaves initially at Arg320 to activate the SP domain (red) and form meizothrombin. The second cleavage event leads to thrombin and F1.2 as products. (**C**) The sequence of the a2-loop of factor fVa is given, with the two acidic regions containing sulfated tyrosines underlined. Residues are numbered, and * denotes sulphation.

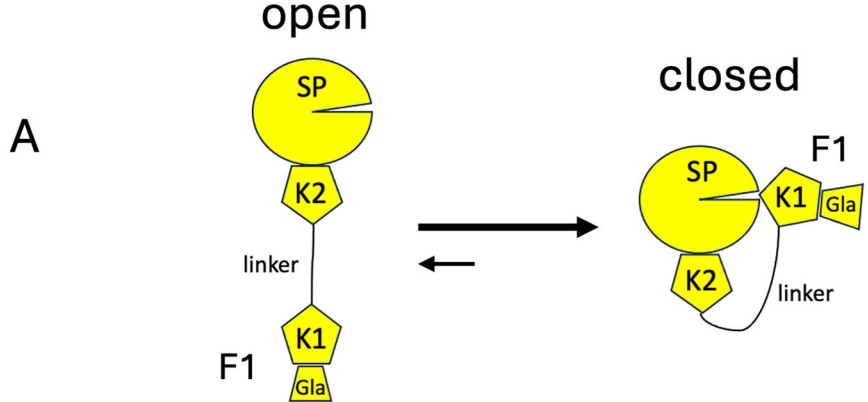

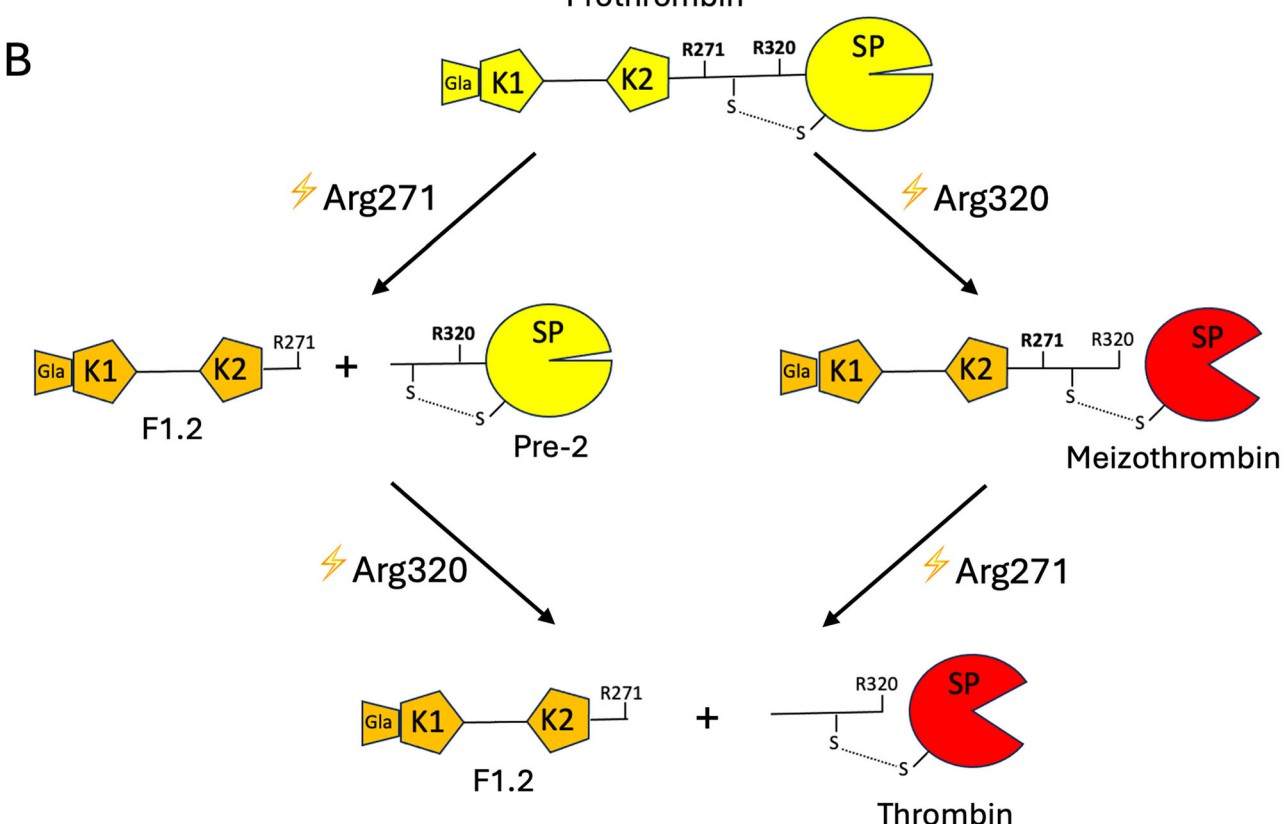

**C**

$_{657}$IP<u>DDDDEDSY*EIFE</u>PPESTVMATRKMHDR<u>LEPEDEESDADY*DY*</u>QNRLAAALGIR$_{709}$

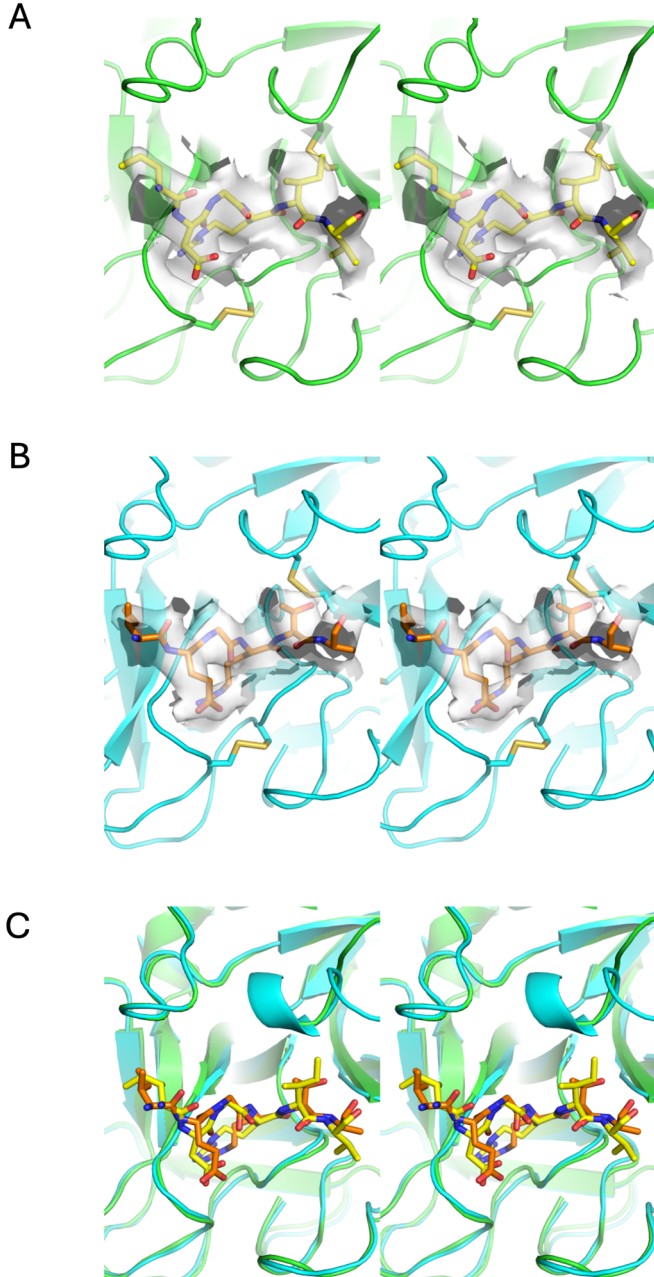

**Figure EV2. Substrate loop interactions within the active site of fXa.**

(**A**) Stereo view of the active site of fXa (green) from the prothrombinase-prothrombin complex, with residues 317-322 of prothrombin (yellow sticks) surrounded by map (semitransparent gray). (**B**) Stereo view of the active site of fXa (cyan) from the prothrombinase-meizothrombin complex, with residues 268-273 of meizothrombin (orange sticks) surrounded by map (semitransparent gray). (**C**) Superposition of the prothrombin and meizothrombin interactions in the active site of fXa, colored as in (**A**, **B**). The figures are in the standard orientation, so that the N-terminal portion of the substrate loop is on the left and the C-terminus on the right.

A

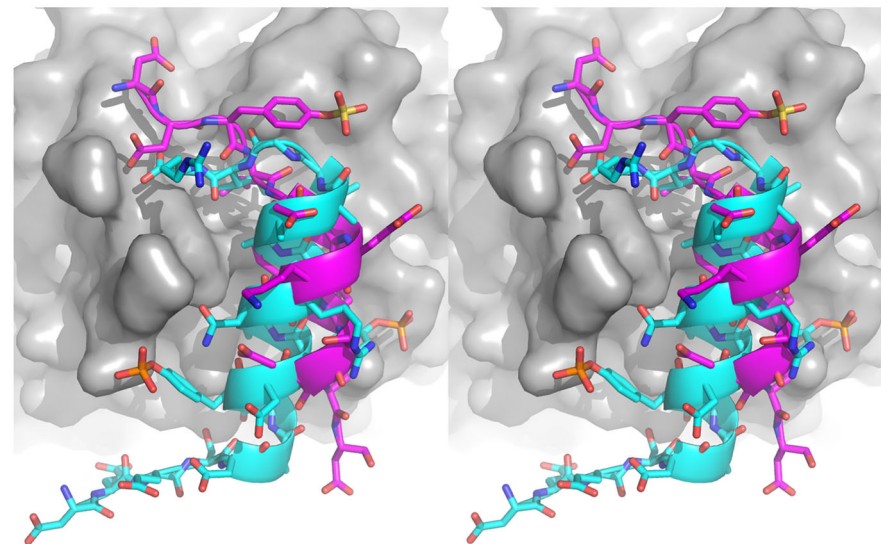

B

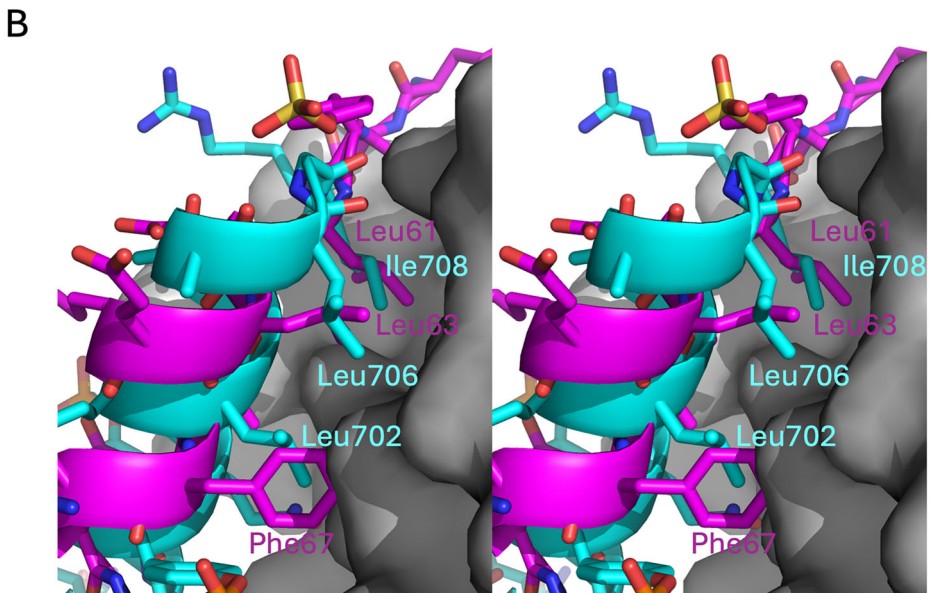

**Figure EV3.  Comparison between a2-loop C-terminal and HCII N-terminal interactions with exosite I.**

(A) Stereo view of the a2-loop (cyan) interaction with prothrombin (gray surface) exosite I in the prothrombinase-prothrombin complex and the interaction of the N-terminal acidic region of heparin cofactor II (HCII; magenta) with exosite I of thrombin. (B) The helices run in opposite directions, but both place key hydrophobic side chains in remarkably similar positions: Phe67, Leu63, and Leu61 for HCII and Leu702, Leu706, and Ile708 for fVa.

Prothrombin          Meizothrombin

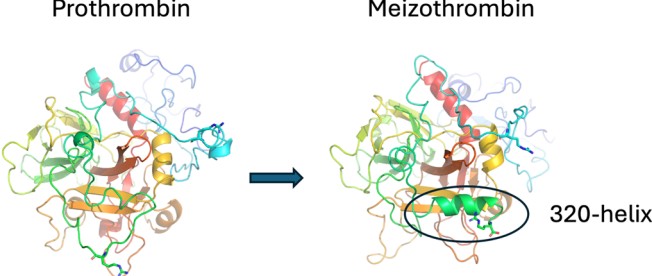

320-helix

**Figure EV4.   Transition of the 320-loop of prothrombin into a helix after cleavage at Arg320 to produce meizothrombin.**

The K2 and SP domains are shown as cartoons, colored from N-to-C-terminus (blue-to-red). The new 320-helix is indicated.

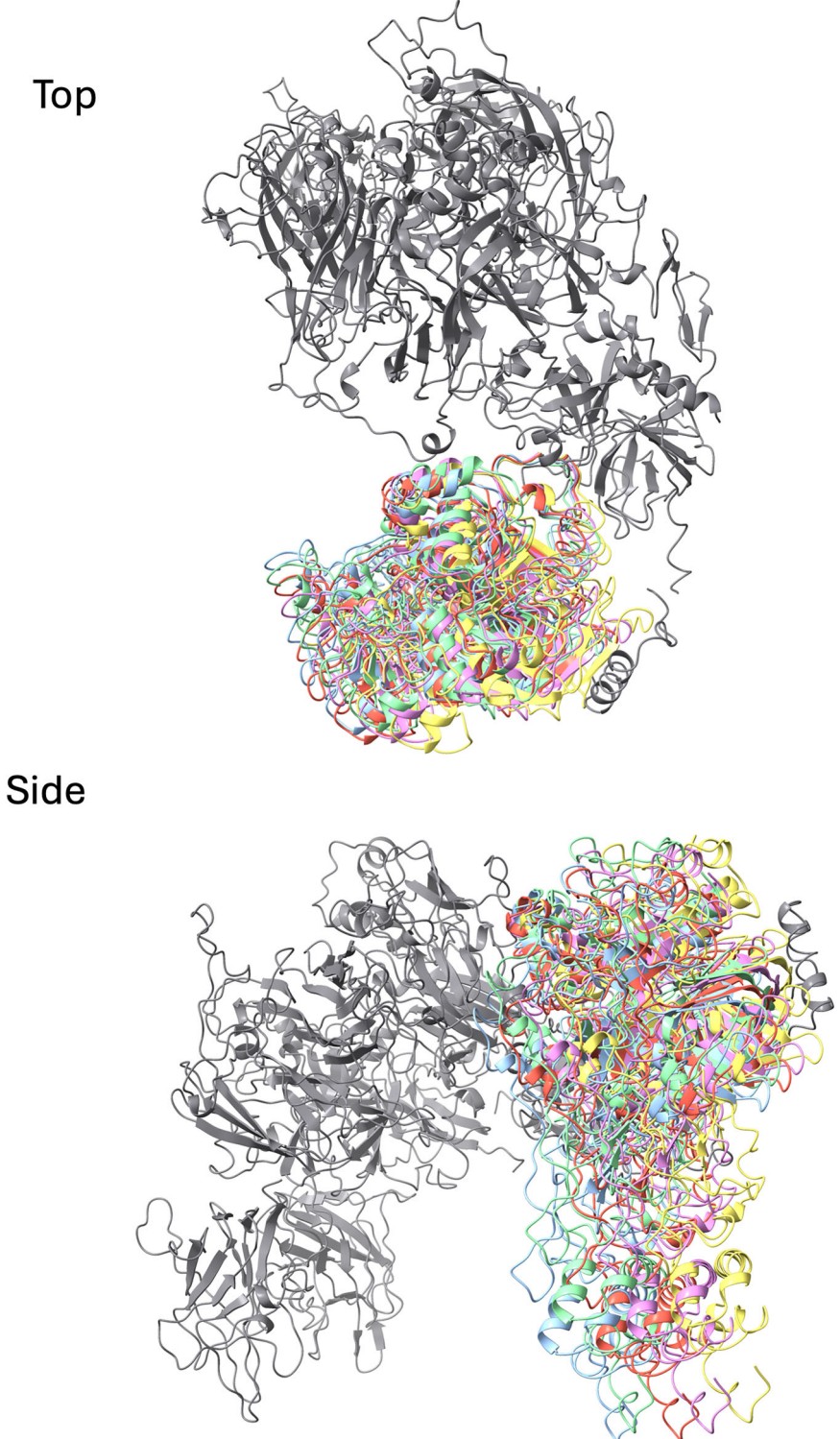

**Figure EV5. Five particle subset classes of the prothrombin component in the prothrombinase complex is illustrated by the fit of prothrombin into the resulting maps.**

Top and side views are shown. Gray is prothrombinase, and the colored cartoons are prothrombin.

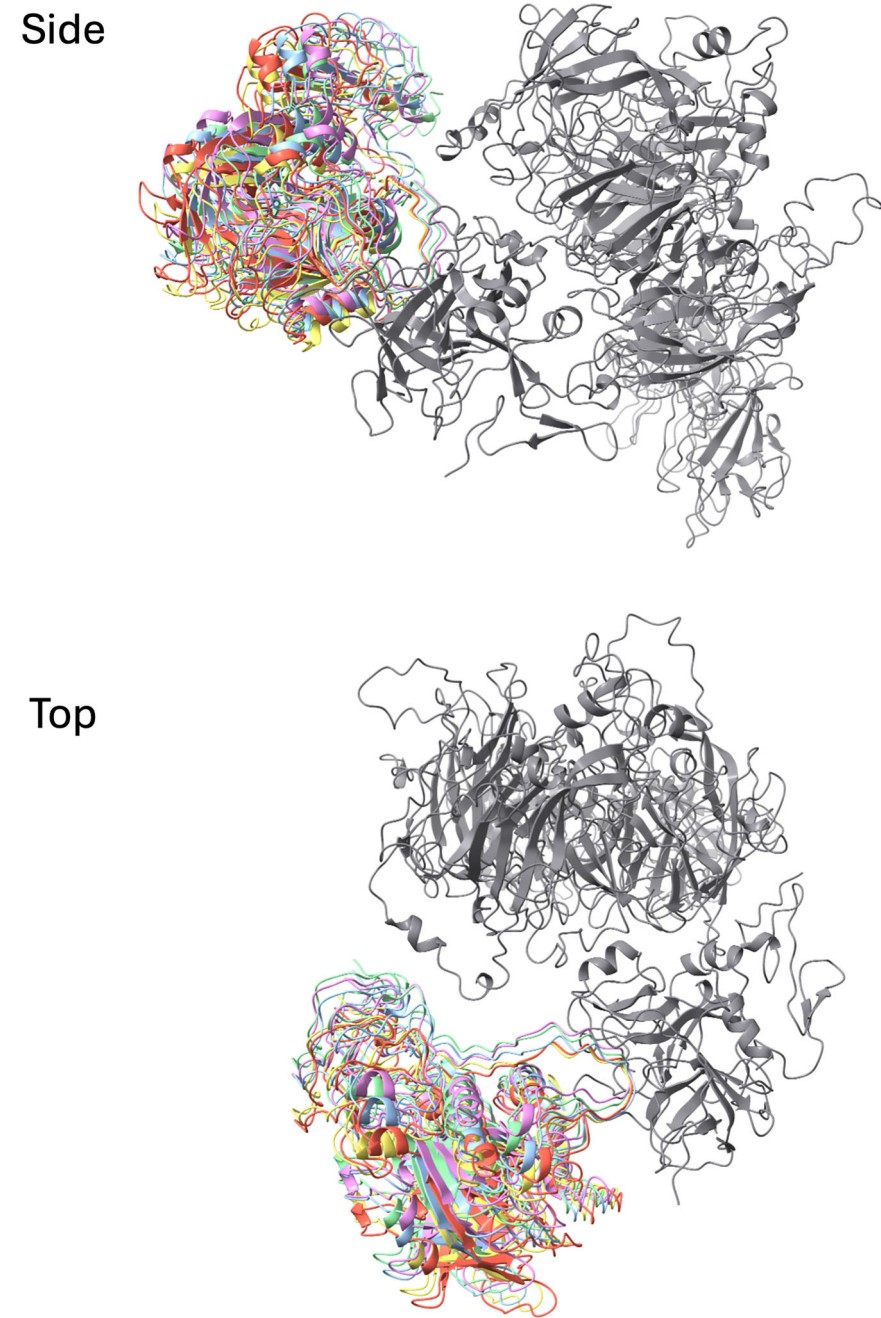

**Figure EV6. Five particle subset classes of the meizothrombin component in the prothrombinase complex is illustrated by the fit of meizothrombin into the resulting maps.**

Side and top views are shown. Gray is prothrombinase, and the colored cartoons are prothrombin.

