## [Peer Review File · The EMBO Journal]

Prothrombinase processivity is conferred by substrate allostery

Fatma Üstok, Alexandre Faille, Alan Warren, and James Huntington

Corresponding author(s): James Huntington (jah52@cam.ac.uk)

Review Timeline:

Submission Date:	13th Feb 26
Editorial Decision:	19th Mar 26
Revision Received:	25th Mar 26
Accepted:	31st Mar 26

Editor: Hartmut Vodermaier

Transaction Report:

Prof. James A. Huntington
Cambridge, University of
Department of Haematology
Cambridge Institute for Medical Research
The Keith Peters Building
Cambridge CB2 0XY
United Kingdom

19th Mar 2026

Re: EMBOJ-2026-123872
Prothrombinase processivity is conferred by substrate allostery

Dear Dr Huntington,

Thank you for submitting your study on prothrombinase processivity for our consideration. I sent it to three expert referees, who have now returned the reports copied below. As you will see, all of them appreciate the importance of the subject and the quality of the work. We shall therefore be happy to publish this study in The EMBO Journal, pending satisfactory responses to a number of specific queries raised in the reviews.

Please keep in mind that it is our policy to consider only a single round of major revision, making it important to adequately answer all comments at the time of resubmission; please do not hesitate to get back to me in case you would like to clarify/discuss any of the referees' points or plans for addressing them ahead of time.

Detailed information on preparing, formatting and uploading a revised manuscript can be found below and in our Guide to Authors, and adhering to them as closely as possible shall greatly facilitate editorial processing upon resubmission. Thank you again for the opportunity to consider this work for The EMBO Journal, and I look forward to your revision in due time.

Yours sincerely,

Hartmut Vodermaier

*** PLEASE NOTE: All revised manuscript are subject to initial checks for completeness and adherence to our formatting guidelines. Revisions may be returned to the authors and delayed in their editorial re-evaluation if they fail to comply to the following requirements. As a first step please read our guidelines for revised submissions:
<https://link.springer.com/journal/44318/submission-guidelines#cms-Revised-submissions>

1) Every manuscript requires a Data Availability section (even if only stating that no deposited datasets are included). Primary datasets or computer code produced in the current study have to be deposited in appropriate public repositories prior to resubmission, and reviewer access details provided in case that public access is not yet allowed.

4) Each main and each Expanded View (EV) figure should be uploaded as individual production-quality files (preferably in .eps, .tif, .jpg formats). For suggestions on figure preparation/layout, please refer to our Figure Preparation Guidelines:
<https://media.springernature.com/original/springer-cms/rest/v1/content/27825798/data/v1>

- 5) Point-by-point response letters should include the original referee comments in full together with your detailed responses to them (and to specific editor requests if applicable), and also be uploaded as editable (e.g., .docx) text files.
- 6) Please complete our Author Checklist, and make sure that information entered into the checklist is also reflected in the manuscript; the checklist will be available to readers as part of the Review Process File.
- 7) All authors listed as (co-)corresponding need to deposit, in their respective author profiles in our submission system, a unique ORCID identifier linked to their name. Please see our Guide to Authors for detailed instructions.
- 8) Please note that supplementary information at EMBO Press has been superseded by the 'Expanded View' for inclusion of additional figures, tables, movies or datasets; with up to five EV Figures being typeset and directly accessible in the HTML version of the article.
- 9) To facilitate reproducibility and cross-laboratory adoption of methodologies, please structure the Materials & Methods section as outlined in our guide to authors, including a completed Reagents and Tools Table.
- 10) Digital image enhancement is acceptable practice, as long as it accurately represents the original data and conforms to community standards. If a figure has been subjected to significant electronic manipulation, this must be clearly noted in the figure legend and/or the 'Materials and Methods' section. The editors reserve the right to request original versions of figures and the original images that were used to assemble the figure. Finally, we generally encourage uploading of numerical as well as gel/blot image source data.

In the interest of ensuring the conceptual advance provided by the work, we recommend submitting a revision within 3 months (17th Jun 2026). Please discuss the revision progress ahead of this time with the editor if you require more time to complete the revisions. Use the link below to submit your revision:

Link Not Available

Referee #1:

The work addresses one of the most fundamental questions covered in every biochemistry textbook. How blood coagulation is regulated and activated by the specific, multistep conversion of prothrombin to thrombin by the prothrombinase complex. The latter consists of the non-enzymatic coagulation factor Va and the activated serine protease coagulation factor Xa. Therefore, it is appropriate to say that the topic of the work is of outstanding importance.

Prothrombin activation requires two consecutive cleavage events: (i) induction of a disorder-to-order transition in the serine protease domain of thrombin, and (ii) release of the serine protease domain from its N-terminal chain. The details of this process are currently poorly understood.

The authors present EM structures at 3.1 Å resolution of the prothrombinase complexed with single-chain prothrombin as well as with two-chain meizothrombin. The latter is a proteolytically activated form in which the serine protease domain has undergone conformational rearrangement to become an active protease, while the two chains remain disulfide-linked. These structures provide important insights into the two-step activation process. The dominant binding interface of prothrombin with factor Va at exosite I is notable. Equally notable are the drastic changes observed in the meizothrombin-factor Va complex interfaces, which emphasise the important role of this highly complex prothrombin substrate.

The manuscript represents an outstanding piece of work. It is very well written, with dedication to detail. I also appreciate the stereo figures with active views in standard orientation. This paper beautifully illustrates why such conventions can be extremely helpful in clarifying interactions in highly complex protein structures.

The structures presented here are highly satisfying and demonstrate the power of structural biology as a whole. The intriguing processivity and efficiency of the prothrombin activation are immediately apparent from the disorder-order transition and the accompanying domain movements induced by a single peptide bond cleavage. As beautiful as it gets.

Minor concerns that should be addressed

Please specify the organism from which the proteins are derived, including the M17 variant. (all human)

please explain the rationale for the stoichiometry 1:6:2

Legend to figure 2: typo: electrostatic surface of prothrombin (m missing)

Which version of AlphaFold was used to predict the interaction of Pre-2 with a2? (AF3?)
In this context: the 5 outputs  the five outputs

Referee #2:

The manuscript by Üstok and colleagues presents structural studies on the activation of prothrombin by its physiological activator, prothrombinase (PT), which in vivo functions exclusively on activated phospholipid (PL) surfaces. Earlier this year (ref. #13), the authors reported the structure of the unbound PT complex, providing data that, in contrast to a previous report from another group, convincingly supported their mechanistic conclusions. To achieve this, they undertook an impressive and technically demanding strategy, engineering 17 mutations within the EGF2 and catalytic domains (CD) to generate a variant (M17) that functions independently of PL surfaces.

The present study advances this work by elucidating the structure of M17 in complex with its substrate in a productive configuration. The authors provide two cryo-EM structures, capturing the successive cleavage events at R320 and R271 through complexes with prothrombin and meizothrombin, respectively. These structures offer important mechanistic insight into the molecular basis of sequential substrate processing by PT.

The study has been conducted with great care and in accordance with state-of-the-art structural and biochemical approaches. The experimental data are of substantial quality and convincingly support the authors' conclusions. The work represents a substantial contribution to the field of coagulation and fibrinolysis, an area to which the Huntington laboratory has made outstanding and sustained contributions over several decades.

I enthusiastically support publication of this manuscript in EMBO Journal.

Minor comments:

- In enzymology, enzymes that require the aid of cofactors such as metal ions or prosthetic groups (e.g. a haeme group, etc.) are holo-enzymes. In the absence of these groups, the enzyme is non-functional and is termed apo-enzyme. A common mistake that can be detected lately in the literature is to call a holo-enzyme that lacks a bound substrate or product an apo-form. Please, replace apo form with unbound form or similar throughout the text.

-It is greatly appreciated that the first five figures, which present the principal findings of the paper, are shown in stereo. This significantly facilitates independent assessment of the structural claims and enhances the clarity of the presentation.

Referee #3:

This study aims to elucidate the structural basis of the prothrombinase complex function and the order of the two-stage processing of prothrombin to thrombin via the meizo-thrombin intermediate. Overall, the manuscript is of very high quality and I find the results and data compelling. One of the surprising findings from the study is the involvement of the A2 loop of FVa with prothrombin. It would be incredibly interesting to determine whether mutation of prothrhombin residues predicted to contribute to this new contact have impaired processivity using human prothrombinase - this could be an excellent means of extrapolating the engineered prothrombinase findings to the human counterpart. For me, this would be the icing on the cake.

A minor comment is the figures. I understand that to those very familiar with the structures of prothrombinase and prothrombin they may be intuitive, but these can take a little deciphering to orient oneself - some additional labelling of the points of interest would appreciably assist the reader in being able to elucidate the features that are being referred to.

We are extremely grateful to the referees for their kind words regarding our manuscript. It is gratifying to have reviewers who understand and appreciate our findings and their implications. Only minor concerns were raised, which I hope are addressed to the referees' satisfaction below and in the edited manuscript.

Referee #1

Point 1: Please specify the organism from which the proteins are derived, including the M17 variant. (all human)

Response: Yes, all human. Now made explicit in the methods section and introduction.

Point 2: Please explain the rationale for the stoichiometry 1:6:2

Response: The 1:6 ratio of fVa to fXa was used in for our structure of substrate-free prothrombinase, and was chosen because it improved orientation distribution at the air-water interface. A 2-fold excess of prothrombin was used just to ensure we would not see substrate-free complex. The small sizes of fXa and prothrombin mean that excess in the grid does not interfere much with the signal contrast.

Point 3: Legend to figure 2: typo: electrostatic surface of prothrombin (m missing)

Response: Fixed. Thanks.

Point 4: Which version of AlphaFold was used to predict the interaction of Pre-2 with a2? (AF3?)

Response: AF3. Now made explicit throughout the text, and the new reference for AF3 replaces the original AF reference.

Point 5: In this context: the 5 outputs  the five outputs

Response: Changed.

Referee #2

Point 1: Please, replace apo form with unbound form or similar throughout the text.

Response: Apo has been replaced with 'substrate-free'.

Referee #3

Point 1: It would be incredibly interesting to determine whether mutation of prothrombin residues predicted to contribute to this new contact have impaired processivity using human prothrombinase - this could be an excellent means of extrapolating the engineered prothrombinase findings to the human counterpart. For me, this would be the icing on the cake.

Response: There is only a small number of prothrombin variants identified in the literature, and although some affect fibrinogen activation and presumably are in exosite I, there is no data (that I can find) regarding the rate of activation by prothrombinase. However, it has been shown that a peptide derived from the C-terminus of hirudin that binds to exosite I of prothrombin inhibits thrombin generation to the same degree as if fVa were not present. It was an oversight not to reference it in this manuscript, which is now remedied in the discussion.

Point 2: A minor comment is the figures. - some additional labelling of the points of interest would appreciably assist the reader.

Response: Labels have been added to Figure 8.